# Emergent Alignment via Competition

Natalie Collina [1]  Surbhi Goel [1]  Aaron Roth [1]  Emily Ryu [2]  Mirah Shi [1]

## Abstract

Aligning AI systems with human values remains a fundamental challenge, but does our inability to create perfectly aligned models preclude obtaining the benefits of alignment? We study a strategic setting where a human user interacts with multiple differently misaligned AI agents, none of which are individually well-aligned. Our key insight is that when the user's utility lies approximately within the convex hull of the agents' utilities, a condition that becomes easier to satisfy as model diversity increases, strategic competition can yield outcomes comparable to interacting with a perfectly aligned model. We model this as a multi-leader Stackelberg game, extending Bayesian persuasion to multi-round conversations between differently informed parties, and prove three results: (1) when perfect alignment would allow the user to learn her Bayes-optimal action, she can also do so in all equilibria under the convex hull condition; (2) under weaker assumptions requiring only approximate utility learning, a non-strategic user employing quantal response achieves near-optimal utility in all equilibria; and (3) when the user selects the best single AI after an evaluation period, equilibrium guarantees remain near-optimal without further distributional assumptions. We complement the theory with two forms of empirical evidence: First, we perform simulations of the best-AI selection game using best response dynamics, which show that competition among individually misaligned agents reliably improves user utility when the approximate convex hull assumption is satisfied, but does not always when it fails. Second, we show that synthetically generated AI utility functions (produced via perturbations of the same prompt

to evaluate instances on a movie recommendation (MovieLens) and ethical judgement (ETHICS) dataset) quickly produce a convex hull that contains a good approximation of a given utility function even when none of the individual LLM utility functions is well aligned. We show that this phenomenon extends to human and LLM responses on real-world polling data (OpinionQA): a convex hull of LLM opinions can approximate human opinions more accurately than any individual LLM across a wide range of survey questions.

## 1. Introduction

Aligning a single AI model to the objectives of its user is a hard problem, not just because of technical complexity, but because the incentives of AI designers may themselves be misaligned with users. But does our inability to solve the alignment problem preclude our ability to get the benefits of interacting with a strong aligned model? In this paper we study a setting in which it does not: when we may interact with multiple *differently misaligned* models in a strategic setting. In particular, we study settings in which there are many AI models available. They are produced by providers like e.g. OpenAI, Anthropic, Google, Meta, AWS, and xAI. These companies produce models reflective of their own incentives, none of which are necessarily well aligned to their user. We note that there has already been significant concern that the designers of LLMs are training them to influence users towards the politics of their creators (Menn, 2025; Kay, 2025; Gilbert, 2024; Hackenburg et al., 2025). In lieu of alignment of any model, we assume instead a much weaker condition on the entire *market* of providers: that (for a well specified task) an approximation of the user's utility function lies somewhere in the *convex hull* of the utility functions of each of the AI companies. This is a condition that does not require that any single model is optimizing a utility function that is similar to that of the human user, and becomes a weaker assumption the more differently aligned models there are that are available to use. We remark at the outset that we primarily use the language of alignment of the human designers, and speak as if these designers are the strategic actors — but we could also think about the agents training and developing these AIs as themselves

[1]Department of Computer and Information Science, University of Pennsylvania [2]Department of Computer Science, Cornell University. Correspondence to: Natalie Collina <ncollina@seas.upenn.edu>, Mirah Shi <mirahshi@seas.upenn.edu>.

*Proceedings of the 43rd International Conference on Machine Learning*, Seoul, South Korea. PMLR 306, 2026. Copyright 2026 by the author(s).

being AIs, whose individual misalignment results from the difficulty of the technical alignment problem. Having AI models themselves involved in the AI training process is a prominent part of thinking about the development of "super-intelligence" (see e.g. (Kokotajlo et al., 2025)) and is already part of current practice in limited ways (Leike et al., 2018; Bai et al., 2022).

There are many ways that our *approximate market alignment* assumption could arise amongst competing AI providers. Consider a near-future scenario in which a human doctor has access to predictive medicine LLMs able to aid in the diagnoses and treatment of patients. The goal of the human doctor might be to provide the best treatment possible for her patients. The LLMs on the other hand might opt for better treatments all else being equal, but might also prefer to prescribe drugs manufactured by a particular company (say if that drug company is the creator or financial sponsor of the model). This preference results in a significantly misaligned model. However, since each drug has a single manufacturer, the "misaligned portion" of the AI model utilities is zero sum, and if all of the relevant drug companies participate in the predictive medicine LLM market, the doctor's utility function will be in the convex hull (in fact the simple average) of the AI model utilities.

Alternately, if the strategic agents are themselves AI models, it may be that their designers attempted to produce them with perfectly aligned utility functions, but failed because the task is difficult. If we view the training of an AI as a stochastic process, we can think of the utility function of an AI model as a random variable whose value is realized during the training process. Perhaps for each AI model, its utility function is — in expectation — equal to the human user's utility function, because that is the target — but its realization has high variance, because alignment is hard. In a setting like this, it may be extremely unlikely that any single trained model is well aligned with the human user, but it will still be very likely that the user's utility function will be close to the convex hull of a large number of trained models because of concentration of measure.

We study how, in settings where approximate market alignment holds, strategic interactions between different models or model providers can allow the human user to realize the full benefit of interacting with a single perfectly aligned model by interacting with many differently misaligned models. While most AI safety research focuses on aligning individual systems or cooperative multi-agent approaches, we study how the benefits of perfect alignment can emerge from market-like competition among misaligned AI providers.

### 1.1. Our Model and Results

We adopt a game theoretic model with Bayesian agents in the style of the Bayesian Persuasion literature (Kamenica & Gentzkow, 2011). A human user named Alice has a set of actions $a \in \mathcal{A}$ that she can take, but which action is best depends on an underlying state of the world $y \in \mathcal{Y}$ that is unknown to her. We model this by endowing Alice with a utility function $u_A : \mathcal{A} \times \mathcal{Y} \to [0,1]$, mapping an action $a$ and a state of the world $y$ to a utility $u_A(a,y)$ that she wishes to maximize. Before taking an action, she can engage in conversation with any of $k$ interlocutors modeling conversational AI agents, all of whom are named Bob. Each Bob $i$ has a (potentially very different) utility function $U_i : \mathcal{A} \times \mathcal{Y} \to [0,1]$ also mapping Alice's action and the state of the world to a utility, which they want to maximize. We assume throughout that Alice's utility approximately lies in the convex hull of the Bob's utility functions (Definition 2.1). There is an underlying prior distribution over triples $x_A, x_B, y$ where $y$ is the state of the world, $x_A$ are observations made by Alice the human user (but possibly not the AI models), and $x_B$ are observations made by the AI models (but possibly not the human user). Alice wishes to converse with the models because the information $x_B$ that they possess is correlated with $y$ and hence potentially decision relevant for her.

The AI designers each commit to a conversation rule, which specifies for any prefix of a conversation how to continue it. This commitment models e.g. fixing the weights of a particular version of an LLM and deploying it. Alice, knowing all of the AI conversation rules, "best responds" with her own conversation rule, and after engaging in conversation with each AI model forms a posterior belief about the state $y$, and then takes the action that maximizes her utility in expectation over this posterior. Thus a set of conversation rules that the AI designers commit to induces through this interaction a joint distribution over outcomes $y$ and actions $a$ that Alice chooses, and gives a different expected utility to each AI. In choosing which conversation rule to commit to, the AI designers find themselves in a simultaneous move game, in which the utility is determined by Alice's downstream use of the deployed models. Our interest is in Alice's utility in the Nash equilibria of this game, played amongst the AI designers.

Our aspirational point of comparison is the utility that Alice could obtain if she were able to interact with a single, perfectly aligned interlocutor. A perfectly aligned provider would choose a conversation rule to maximize *Alice*'s utility after she best responded (i.e. used the model optimally). Our results explore settings in which this goal is obtainable even when none of Alice's interlocutors are individually well aligned, in increasing order of generality. In all of the following results we assume that Alice's utility approximately lies in the convex hull of each of the AI model's utility functions (or more generally is a non-negative linear combination of them).

1. First we show in Section 3 that whenever it is feasible for a single model to engage in a conversation with Alice that causes her to learn her Bayes optimal action $a^* = \arg\max_{a \in \mathcal{A}} \mathbb{E}[u(a, y)|x_A, x_B]$ — and hence, whenever a perfectly aligned model would cause Alice to do so, then if Alice's utility function lies in the convex hull of the Bob's utility functions, in any Nash equilibrium of the game, Alice is able to learn her Bayes optimal action — and hence do as well as if she were interacting with a perfectly aligned model.

2. In Section G we study a model in which Alice acts non-strategically: she always interacts with AIs using a *straightforward* conversation rule, which truthfully reports the posterior expectation of each of her actions at each round of conversation. At the end of conversation, she chooses her action using quantal response (a form of "smooth best response" in which the maximum is replaced by a softmax operator, which is a common model of bounded rationality in the behavioral economics literature (McKelvey & Palfrey, 1995)). We can view these assumptions either as modeling a boundedly rational Alice (as they would be interpreted in the behavioral economics literature), or as explicit behavioral commitments that a strategic Alice makes in order to be able to enjoy the more robust guarantee that we prove under this model. In particular we can relax the condition that Alice is able to learn her Bayes optimal action exactly when conversing with a perfectly aligned model to the condition that she learns the *approximate* utility of playing each of her actions — i.e. she is able to approximate $\mathbb{E}[u(a, y)|x_A, x_B]$ for each $a$. We show that this weaker condition suffices for Alice to obtain (approximately) the utility that she could have obtained interacting with a perfectly aligned model in every Nash equilibrium of the game induced amongst the AI models. In particular, if the underlying distribution satisfies the "information-substitutes" condition studied by (Frongillo et al., 2021) or its generalization studied by (Collina et al., 2025a), we show that this is enough to guarantee that a perfectly aligned model could inform Alice of the approximate Bayes utilities of each of her actions, allowing us to invoke our equilibrium guarantees.

3. In Section H we dispense with all assumptions on the distribution and instead change the communication protocol. Rather than assuming that Alice will interact with *all* $k$ of the AI models before making each decision, we assume that once the $k$ AI providers commit to a set of conversation rules, Alice will evaluate each of them to compute the expected utility (over the distribution of instances) that she would get by interacting with each one individually, and then will choose to interact with only the single model that guarantees her highest expected utility, for all instances. We can view this either as a behavioral commitment on Alice's part or a modeling assumption about the market (i.e. maybe Alice signs a contract with only one of the model providers after an evaluation period). In this case, we show that without any further assumptions on the instance, in equilibrium Alice is always able to obtain utility comparable to what she could have obtained by interacting with a perfectly aligned model.

4. In Section J we conduct a simple stylized experiment designed to test our core premise that given a set of AI models, there may be a utility function in the convex hull of the set of all AI agent utility functions that is substantially better aligned than any of the individual AI utility functions themselves. We test this premise in experiments on a few datasets. In the first we simulate a "human" utility function by using an LLM with a hand-crafted prompt, and ask it to evaluate 1000 ethical scenarios from the ETHICS dataset (Hendrycks et al., 2021). To simulate "AIs" that are designed to be aligned with the human utility function but are only noisy approximations, we produce perturbations of the original ("human") prompt by asking a language model to rephrase the prompt while maintaining its core intent. We produce 100 such perturbations, resulting in up to 100 "AI personas" that we also use to evaluate the same 1000 ethical scenarios. Finally as a function of the number of AI models $k$ we evaluate the alignment of 1) the best aligned of the $k$ AI personas and 2) the best aligned utility function that can be computed within the convex hull of the $k$ AI personas. We repeat the experiment on the MovieLens dataset (Harper & Konstan, 2015) in which we use the average human annotation of movies as the "human" utility and similarly simulate 100 AI utility functions through 100 variations of a prompt. Finally, we move to real human preference data, using an existing human-LLM polling dataset of public opinion survey responses (Santurkar et al., 2023). Here, we treat individual human responses as human utilities and LLM responses as AI utilities. On all datasets we find that the best utility function in the convex hull of the AI utility functions is substantially better aligned to the "human" than any of the AI personas themselves. This supports our main conceptual contention that the target of alignment within the convex hull of many models may be substantially easier to obtain than alignment of any single model individually. Finally, we conduct a simple stylized experimental evaluation of the equilibria of a variant of the game we study in Section H. We compute equilibria of a tractable special case of this game with a variety of utility functions, and find that consistent with our theory, Alice's utility in equilibrium is always at least as high as predicted by our theory based on the alignment error of the best approximation to her utility function in the convex hull of the AI providers utilities —

and sometimes substantially better.

## 1.2. Related Work

**Bayesian Persuasion**  Bayesian Persuasion was introduced by (Kamenica & Gentzkow, 2011) — in the canonical model, there is a single informed "sender" and an uninformed "receiver" who share a common prior. The sender commits to a "signaling scheme", which is a mapping from observations to messages sent to the receiver, who conditions on the message and takes their best response action under their posterior. We adopt the basics of this model, but extend it by allowing that both parties be differently informed, and that interaction involve a multi-round conversation rather than a single message. Multi-sender Bayesian Persuasion was introduced by (Gentzkow & Kamenica, 2017a) and studies the standard Bayesian Persuasion model with multiple senders who simultaneously commit to a signaling scheme (playing, as in our paper, a simultaneous move commitment game). A number of papers have since studied multi-sender Bayesian Persuasion (Gentzkow & Kamenica, 2017b; Li & Norman, 2018; Au & Kawai, 2020; Wu, 2023). We focus here on the most relevant papers.

(Ravindran & Cui, 2020) study competing senders with zero-sum preferences over a receiver's beliefs. They show that competition leads to full revelation of the state in all equilibria, provided the senders' utility functions are "globally nonlinear". This technical condition can hold in a standard receiver model only if the receiver has a different optimal action for every distinct state of the world. This condition cannot hold whenever e.g. the number of states of the world exceeds the number of actions. Our work does not assume that the leaders Bob are engaged in a zero-sum game with each other — rather our market alignment assumption can be viewed as assuming that the misaligned portions of their utility functions are approximately zero-sum under some non-negative reweighting. We also do not require an analogue of the "globally nonlinear" assumption, and so our results can apply to settings with large state spaces.

**AI Alignment**  Our work fits broadly into the study of multi-agent AI systems (Guo et al., 2024). We present a game theoretic approach in which "alignment" emerges from the competitive interaction of many mis-aligned agents. Recent work has explored cooperative multi-agent approaches to AI safety, where multiple AI systems work together to improve alignment outcomes. Constitutional AI (Bai et al., 2022) uses AI feedback to train more helpful and harmless models, with one AI system providing critiques and revisions of another's outputs. Similarly, approaches using AI systems to evaluate and improve other AI systems (Leike et al., 2018) rely on cooperative dynamics where the evaluating system is assumed to be sufficiently aligned to provide useful feedback. These approaches typically assume that at least some components of the multi-agent system are well-aligned or that the agents share compatible objectives. Our work differs by studying strategic rather than cooperative multi-agent settings.

Several recent papers with alignment motivations (Collina et al., 2025b;a; Nayebi, 2025) have studied *agreement protocols* through which conversational agents can come to agreement about their beliefs through short interactions. These should be viewed as protocols for cooperative agents, as they are assumed to express their true beliefs at each iteration of conversation. We adopt the conversational framework of these papers but study *strategic* agents who do not have the same goals. Our work can both be viewed as a strategic generalization of the agreement literature (Aaronson, 2005; Frongillo et al., 2021; Collina et al., 2025a;b; Nayebi, 2025), and a generalization of the (already strategic) Bayesian Persuasion literature beyond simple one-round signaling schemes used to communicate between an informed party and an uninformed party to multi-round conversation protocols used by differently informed parties.

In terms of techniques, the most closely related paper is the concurrent work of (Fudenberg & Liang, 2025) who study the interaction of a risk-averse user with a single (potentially misaligned) AI, which they model as a principal agent game. They ask the question of how much information they should reveal to the AI in the best case (in which the AI is perfectly aligned, and nature produces the best possible outcome), and in the worst case (in which nature and a misaligned AI collude to produce the worst-case outcome), and study the Pareto frontier of their information design problem. We defer further discussion of related work to Section A.

## 2. Preliminaries

This section establishes the formal framework for our analysis. We first introduce the players and their information structure, then present our key modeling assumption about approximate market alignment, and finally define the communication protocol and game structure.

**Players and Information Structure.**  We model the interaction as a multi-leader Stackelberg game, extending the Bayesian persuasion framework to our setting. We model AI providers as "leaders" who commit to conversation strategies first, knowing that the human user (follower) will observe these strategies and respond optimally. This captures the reality that AI systems are deployed with fixed parameters, while users can adapt their interaction strategies. Alice observes features $x_A \in \mathcal{X}_A$ and must choose an action $a \in \mathcal{A}$. Each $\text{Bob}_i$ observes features $x_B \in \mathcal{X}_B$. There is a state of the world $y \in \mathcal{Y}$ that is not directly observed by any player. All players have utility functions that depend on Alice's action and the state of the world to a utility in $[0, 1]$:

$u_A : \mathcal{A} \times \mathcal{Y} \to [0, 1]$ and $U_i : \mathcal{A} \times \mathcal{Y} \to [0, 1] \quad \forall i \in [k]$.

**The Market Alignment Assumption.** We now turn to our key modeling assumption: that Alice's utility can be approximately represented as a weighted combination of the AIs' utilities, a notion we call "market alignment."

**Definition 2.1** (Approximate Market Alignment). *A key assumption of our model is that there exists a weighted combination of the Bobs' utilities that is approximately aligned with Alice's. Formally, we assume there exist non-negative weights $w_1, \ldots, w_k \geq 0$ with $\sum_i w_i = 1$, an offset $c \in \mathbb{R}$, and an alignment error $\varepsilon \geq 0$ such that:*

$$\sup_{a \in \mathcal{A}, y \in \mathcal{Y}} \left| \left( \sum_{i=1}^{k} w_i U_i(a, y) + c \right) - u_A(a, y) \right| \leq \varepsilon.$$

*This assumption is central to our results.*

**Communication Protocol and Game Structure.** With this in place, we can now define the communication protocol that governs how Alice and the AIs interact.

**Probabilistic Model and Beliefs.** We assume there is a commonly known prior distribution $P(x_A, x_B, y)$ over Alice's features, the Bobs' features, and the state of the world. Given some information $\mathcal{F}$ (e.g., a conversation transcript or a subset of features), Alice forms a belief about her expected utility for each action. We denote this belief vector as $\mu(\mathcal{F}) := (\mathbb{E}_y[u_A(a, y) \mid \mathcal{F}])_{a \in \mathcal{A}}$.

**Definition 2.2** (First-Best Utility). *We define the first-best utility, $OPT$, as Alice's expected utility if she had access to all features $(x_A, x_B)$. Formally:*

$$OPT := \mathbb{E}_{(x_A, x_B)} \left[ \max_{a \in \mathcal{A}} \mathbb{E}_y[u_A(a, y) \mid x_A, x_B] \right].$$

**Remark 2.3.** *The first-best utility $OPT$ represents Alice's utility if she had perfect information—knowing both her private features $x_A$ and the AIs' private features $x_B$. This serves as an upper bound on what any communication protocol can achieve, since no amount of conversation can provide Alice with more information than she would have with direct access to all features.*

**The Communication Protocol.** The communication protocol models realistic constraints on human-AI interaction: conversations have limited rounds, messages have bounded complexity, and the human must process information from multiple AIs simultaneously. Alice engages in parallel private conversations with each AI, which captures settings where she can query multiple models independently. In most of the paper, Alice engages in $R$ rounds of private, parallel conversations with each of the $k$ Bobs (we will change the protocol in Section H). Let $M$ be the message space.

We now formalize each player's strategic choices. Each AI commits to a conversation rule (how to respond given the conversation history) while Alice chooses both a conversation rule (how to query the AIs) and a decision rule (how to act given the final conversation outcomes).

**Definition 2.4** (Player Strategies). *Each player's strategy is defined by a set of rules governing their communication and decisions.*

- *Bob$_i$'s **conversation rule** $C_{B_i} : \mathcal{X}_B \times M^{<R} \to \Delta(M)$. maps his features and his private conversation history with Alice to a distribution over messages:*

- *Alice's **conversation rule** $C_A : \mathcal{X}_A \times (M^{<R})^k \to \Delta(M^k)$. maps her features and the full history of all $k$ conversations to a distribution over next messages for each Bob:*

- *Alice's **decision rule** $D_A : \mathcal{X}_A \times (M^R)^k \to \Delta(\mathcal{A})$. maps her features and the full conversation history to a distribution over actions:*

**Definition 2.5** (Best Response Decision Rule). *A **best-response decision rule** is a deterministic rule $D_A^*$ that, given the final posterior belief $\mu(x_A, \pi)$ derived from Alice's features $x_A$ and a transcript $\pi$, selects an action that maximizes Alice's expected utility:*

$$D_A^*(x_A, \pi) \in \arg\max_{a \in \mathcal{A}} \mu_a(x_A, \pi).$$

*In cases of ties, a fixed, predetermined rule is used.*

**Conversation Rule Examples.** For the Bobs, conversation rules correspond to the deployed model policy: for example, "Model X with provider Y's safety layer and refusal policy." Alice observes this conversation rule indirectly by knowing which API or product she is using, and how it has responded in the past to her and to others.

For Alice, a conversation rule is her repeated interaction strategy with the models. This interaction strategy may be adaptive to new information, not only within a conversation with one model, but across conversations. Examples include:

- "Ask all models the same question."

- "Ask all models the same question, then pick the model with the promising answer and ask adaptive follow-up questions only to that model."

- "Ask all models the same question, then show the most promising answer to the other models and ask for critiques."

- "Phrase the question in $k$ different ways and ask each of the models a differently-phrased question."

**The Game.** The game proceeds as a multi-leader, single-follower Stackelberg game, with the following timing:

1. Each $Bob_i$ simultaneously commits to a conversation rule $C_{B_i}$.

2. Alice observes the chosen conversation rules $\vec{C}_B = (C_{B_1}, \ldots, C_{B_k})$, and then chooses her own conversation rule and decision rule $C_A$ and $D_A$.

3. An instance $(x_A, x_B, y)$ is sampled from the prior distribution $P$. Alice observes $x_A$ and each Bob observes $x_B$.

4. Alice and the Bobs engage in the communication protocol defined by $\vec{C}_B$ and Alice's own conversation rule $C_A$ to sample a conversation transcript $\pi$. The protocol is defined precisely in Algorithm 1.

5. Alice samples an action $a$ according to her decision rule $a = D_A(x_A, \pi)$, and all players receive their utilities $u_A(a, y)$ and $U_i(a, y)$.

**Induced Distributions and Equilibria.**

**Definition 2.6** (Induced Distribution). *A set of strategies $(\vec{C}_B, C_A, D_A)$ induces a joint distribution over conversation transcripts $\pi$, actions $a$, and world states $y$. We denote the marginal distribution over actions and outcomes by $\mathcal{I}(\vec{C}_B, C_A, D_A)$.*

Since Alice observes the Bobs' conversation rules $\vec{C}_B$ before choosing her own, she will play a best response. A rational Alice will always use the **Best Response Decision Rule** (Theorem 2.5) to select her action after the conversation concludes. Therefore, her only strategic choice is her conversation rule, $C_A$.

**Definition 2.7** (Alice's Best-Response Conversation Rule). *Given a vector of Bobs' conversation rules $\vec{C}_B$, Alice's best-response conversation rule $C_A^*$ is one that maximizes her expected utility, assuming she will use the best-response decision rule $D_A^*$:*

$$C_A^* \in \arg\max_{C_A} \mathbb{E}_{(a,y) \sim \mathcal{I}(\vec{C}_B, C_A, D_A^*)}[u_A(a, y)].$$

*When multiple conversation rules yield the same maximal utility, a fixed tie-breaking rule is used. We write $C_A^* = C_A^*(\vec{C}_B)$ to make the dependency on $\vec{C}_B$ explicit.*

Since Alice plays a best response, we can define the resulting induced distribution as a function of the Bobs' strategies alone: $\mathcal{I}^*(\vec{C}_B) = \mathcal{I}(\vec{C}_B, C_A^*(\vec{C}_B), D_A^*(\vec{C}_B))$. With Alice's response fixed, the Bobs engage in a simultaneous-move game. We study the Nash equilibria of this game.

**Definition 2.8** (Nash Equilibrium). *A vector of Bobs' conversation rules $\vec{C}_B^* = (C_{B_1}^*, \ldots, C_{B_k}^*)$ is a Nash Equilibrium if no $Bob_i$ can improve his expected utility by unilaterally deviating to a different rule $C'_{B_i}$. That is, for all $i \in [k]$ and for all alternative rules $C'_{B_i}$:*

$$\mathbb{E}_{(a,y) \sim \mathcal{I}^*(\vec{C}_B^*)}[U_i(a, y)] \geq \mathbb{E}_{(a,y) \sim \mathcal{I}^*((C'_{B_i}, \vec{C}_{B, -i}^*))}[U_i(a, y)].$$

Nash equilibria can be shown to exist in our setting under the same conditions under which they are known to exist in multi-sender Bayesian Persuasion (Gentzkow & Kamenica, 2017a;b; Hossain et al., 2024). For more details, see Appendix C.1.

Our interest is in lower bounding Alice's utility in *all* Nash equilibria of this game. In particular, we will be interested in settings in which her utility is guaranteed to be competitive with what she would have received were Alice to be interacting with a single, perfectly aligned leader.

**Definition 2.9** (Utility with an Aligned Leader). *A useful benchmark is the utility Alice could achieve if she were interacting with a single, perfectly aligned leader Bob. A perfectly aligned leader is one whose utility function is identical to Alice's, i.e., $U_B(a, y) = u_A(a, y)$. Such a leader would choose a conversation rule $C_B^*$ to maximize Alice's expected utility. We denote this maximum achievable utility as $U_A(C_B^*)$:*

$$U_A(C_B^*) := \max_{C_B} \mathbb{E}_{(a,y) \sim \mathcal{I}^*(C_B)}[u_A(a, y)].$$

*This represents the best possible outcome for Alice given the constraints of the communication protocol with a single, fully cooperative partner.*

Note that the utility that Alice can obtain when interacting with a perfectly aligned leader is at most her first best utility: $U_A(C_B^*) \leq OPT$. In some situations we will have $U_A(C_B^*) = OPT$ (for example if the message space is sufficiently expressive to encode $x_A$ over $R$ rounds of communication), but if the message space is more restrictive the inequality could be strict.

## 3. Competition Achieves Optimal Outcomes in Ideal Scenarios

Our first result shows that if Alice could achieve her first-best utility by talking to a single perfectly aligned AI, then she can achieve nearly the same utility in equilibrium when talking to many misaligned AIs—provided her utility lies in the convex hull of theirs.

This section establishes this result through two steps. First, we identify a key structural condition—the "Identical Induced Distribution Condition"—that captures when there is a fixed deviation such that different Bobs adopting the same

deviation lead to the same decisions by Alice (Section 3.1) — i.e. Alice's behavior depends on what she learns, but not who taught it to her. Second, we prove that under this condition, strategic competition automatically leads Alice to achieve near-optimal utility (Section 3.2). We observe that this condition is in particular satisfied when a perfectly aligned Bob could cause Alice to learn her Bayes optimal action. Full proofs are provided in Appendix E and F.

### 3.1. The Identical Induced Distribution Condition

The key technical condition driving our result is that it "doesn't matter" which Bob adopts the Alice-optimal strategy—Alice gets the same outcome regardless. This holds, for example, when the Alice-optimal strategy allows her to learn her Bayes-optimal action, since she'll act on this no matter who teaches it to her.

We now formalize this condition. Let $C_B^*$ be a conversation rule for a single leader that maximizes Alice's utility (i.e. a conversation rule that a perfectly aligned Bob would use), and let $U_A(C_B^*)$ be this maximum single-leader utility.

**Definition 3.1** (Identical Induced Distribution Condition). *A game structure satisfies the identical induced distribution condition if for any strategy profile $\vec{C_B}$ and any two Bobs $i, j \in [k]$, the distributions induced by a unilateral deviation to $C_B^*$ are identical. That is,*

$$\mathcal{I}^*((\vec{C_B}^{-i}, C_B^*)) = \mathcal{I}^*((\vec{C_B}^{-j}, C_B^*)).$$

Here, $\vec{C_B}^{-i}$ denotes the vector of all other Bobs' strategies, and $\mathcal{I}^*((\vec{C_B}^{-i}, C_B^*))$ is the induced distribution when Bob $i$ unilaterally deviates to $C_B^*$.

Observe that the Identical Induced Distribution Condition will hold in any setting in which it is in Bob's strategy space to cause Alice to learn her optimal action (and hence obtain her first-best utility $OPT$):

**Proposition 3.2** (When the Condition is Satisfied). *The identical induced distribution condition is satisfied if the Alice-optimal leader strategy $C_B^*$ allows Alice to learn her Bayes-optimal action $a^*(x_A, x_B) = \arg\max_{a \in \mathcal{A}} \mathbb{E}_y[u_A(a, y)|x_A, x_B]$.*

Note that Proposition 3.2 is a joint condition on Alice's utility and the expressiveness of the Bobs' message spaces, and is fully independent of the Bobs' utilities.

**Remark 3.3.** *A straightforward case where the condition of the proposition holds is when the message space $M$ is rich enough to contain the Bobs' feature space $\mathcal{X}_B$. In this setting, an optimal strategy $C_B^*$ can be for the Bob to simply reveal $x_B$ to Alice. With full knowledge of $(x_A, x_B)$, Alice can compute her Bayes-optimal action $a^*(x_A, x_B)$.*

Having established when the identical induced distribution condition holds, we now show that this condition is sufficient to guarantee that Alice achieves near-optimal utility in equilibrium. The proof relies on a simple observation about Nash equilibria: no Bob wants to deviate—in particular, to any conversation rule that would make Alice better off—but this constraint, combined with our alignment assumption, forces Alice's utility to be high.

### 3.2. Strategic Competition Leads to Near-Optimal Outcomes

We can now state our first result: under the identical induced distribution assumption, approximate market alignment implies that in equilibrium, Alice gets utility approximately that of interacting with a single, perfectly aligned leader. In particular, if the message space is expressive enough to allow an aligned leader to communicate to Alice her Bayes-optimal action, approximate market alignment is sufficient for Alice to obtain approximately her first-best utility.

**Theorem 3.4.** *If the multi-leader game satisfies the identical induced distribution condition, and if the Bobs satisfy the $\varepsilon$-market alignment condition, then Alice's expected utility in any Nash equilibrium is at least $U_A(C_B^*) - 2\varepsilon$.*

### 3.3. Extensions

The previous result provides strong guarantees but requires the identical induced distribution condition to be satisfied. In Section G, we show an approximate version of this condition (Definition G.6) which we use in this section. While the original condition is satisfied when Alice is able to learn her exact Bayes optimal utility, we show that the relaxed condition is satisfied when Alice is able to learn it *approximately*.

To analyze this weaker setting, we study a model where Alice acts straightforwardly rather than strategically, committing to two behavioral rules: (1) she always reports her honest beliefs during conversation, and (2) she uses "quantal response" for decision-making—a form of bounded rationality where she chooses actions probabilistically based on their estimated utilities rather than always picking the best one. We can view these assumptions either as modeling a boundedly rational Alice, or as explicit behavioral commitments that a strategic Alice makes to enjoy more robust guarantees. In particular, if the information structure satisfies an "information substitutes" condition (Definition G.11), then Alice's utility with an aligned leader is in fact her first-best utility $OPT$.

**Theorem G.15** (Informal). Suppose the underlying distribution satisfies the Information Substitutes Condition (Definition G.11) and the $\varepsilon$-market alignment condition. If Alice commits to the straightforward conversation rule and a $\lambda$-

quantal response decision rule, her expected utility in any Quantal Response Nash Equilibrium of the induced game is close to the first-best optimal utility $OPT$.

Then in Section H, we give a modified game in which Alice is guaranteed approximately the utility she could get by interacting with a single perfectly aligned AI, without any variant of the identical induced distribution condition. To do this, we modify the design of the game. In the interaction we study now, the $k$ leaders Bob still commit to conversation rules. But now, rather than interacting with all $k$ of these conversation rules at decision time, Alice (after observing the $k$ conversation rules deployed by the Bobs) chooses one to interact with — i.e. the one that guarantees her the highest expected utility over the prior distribution. She then deploys a best-response conversation and decision rule to interact with only this single conversation rule. We can view this either as a behavioral commitment on Alice's part (to enjoy the more robust guarantees that we prove in this Section), or a model of existing practice — that e.g. Alice or her employer might, after a period of evaluation, contract with just a single LLM provider. We show that the market alignment assumption is enough to guarantee that Alice does as well in the equilibrium of this game as she would interacting with a perfectly aligned single model Bob.

**Theorem H.3** (Informal). *Consider a Best-AI Selection game such that the $\varepsilon$-market alignment condition is satisfied. In any Nash Equilibrium, Alice's expected utility is at least $U_A(C_B^*) - 2\varepsilon$, where $U_A(C_B^*)$ is Alice's utility with an aligned Bob.*

## 4. Experiments

We provide two complementary empirical investigations of our theoretical results. First, we test our key assumption that Alice's utility can be well-approximated within the convex hull of misaligned AI utilities, even when no individual AI is well-aligned. Second, we simulate equilibrium outcomes in the Best-AI Selection Game from Section H to verify that competition leads to high utility for Alice.

### 4.1. Market Alignment Experiments

**Setup.** We test whether Alice's utility can be recovered as a non-negative weighted combination of misaligned Bobs. First, we use real-world polling data from the OpinionQA dataset (Santurkar et al., 2023), which contains LLM (Bobs) and human (Alice) responses on survey questions spanning cultural and political issues. We also test the same hypothesis on ETHICS (Hendrycks et al., 2021)—for moral judgments—and MovieLens (Harper & Konstan, 2015)—for movie recommendations—by generating 100 diverse Bobs via LLM prompt variations. Across all three domains, we measure how well Alice's utility can be reconstructed

using non-negative weighted combinations as the number of available Bobs increases.

**Results.** Figure 1 compares alignment error of the best non-negative weighted combination of Bobs' utilities (NNLS), the best convex combination (simplex), the average with equal weights (simple average), and the utility of the single Bob closest to Alice's (best individual). Figure 1a shows that weighted combinations of model opinions is substantially better aligned with human opinions than any individual model, and alignment error decreases as more models are added. Figure 1b shows a similar finding for MovieLens. Here, at $K = 100$, non-negative combinations reduce error by $70 - 75\%$ relative to the best individual Bob, while simple averaging performs poorly. This supports our key market alignment assumption.

### 4.2. Strategic Equilibrium Experiments

**Setup.** We compute equilibria in simplified settings (synthetically generated utility tables consisting of 3-5 states, 3-9 actions, 5-6 Bobs, plus MovieLens-derived utilities), using best-response dynamics and enumeration. We compare Alice's realized equilibrium utility to our theoretical bounds, and also evaluate a *misalignment score* to measure the degree of violation to our market alignment condition (essentially a computation of $\varepsilon$ in the condition).

**Results.** Figure 1c (MovieLens) shows that Alice's utility in equilibrium consistently meets or exceeds our theoretical bound of $OPT - 2\varepsilon$; this holds across all tested scenarios. Lower misalignment scores strongly predict higher equilibrium utility for Alice, and competition among even small numbers of Bobs (2-3) provides significant benefits over single-Bob interactions.

Together, these results demonstrate that (i) diverse misaligned Bobs can collectively approximate Alice's preferences, and (ii) strategic competition among them guarantees her high equilibrium utility. Full experimental details and additional results are provided in Appendix J, K, and L.

We have introduced a new approach to AI alignment through competition between multiple, differently misaligned models so that the benefits of perfect alignment emerge in equilibrium. The key condition is that the user's utility function can be approximately represented as a non-negative weighted combination of the AI providers' utility functions, which is much more robust than requiring any single AI provider to be perfectly aligned. We view this work as the first step in a broader research agenda of mechanism design for AI alignment, with future directions including developing protocols with more robust guarantees and extending to diverse populations of downstream users. Our model suggests important questions about testing existing model

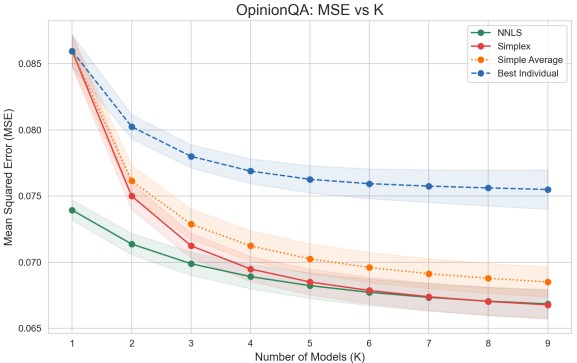

*(a)* Alignment error vs number of Bobs (Opinion QA)

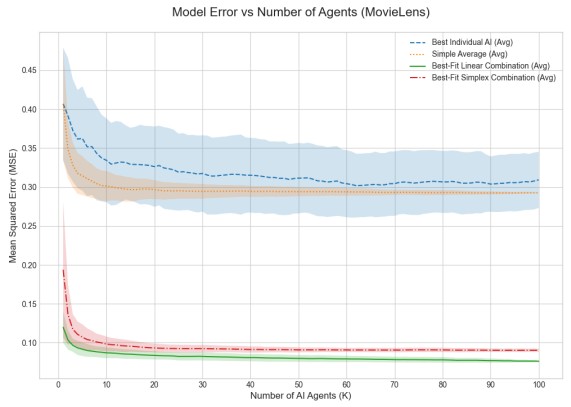

*(b)* Alignment error vs number of Bobs

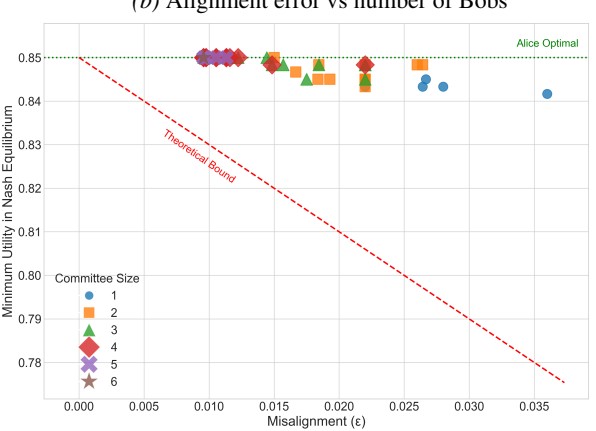

*(c)* Misalignment ($\varepsilon$) vs minimum Alice equilibrium utility

*Figure 1.* Experimental results for MovieLens dataset. *(a)* Alignment error (MSE) decreases as more agents are added to the convex hull. Weighted combinations (NNLS in green, simplex in red) substantially outperform both the best individual agent (blue) and simple average (orange), with error dropping by 50-70% at $K = 100$. Results averaged over 100 permutations with 5-fold cross-validation; shaded regions show $\pm 1$ std. dev. *(b)* Marker shape encodes committee size $k$. Dashed red: $OPT - 2\varepsilon$. Dotted green: Alice-optimal utility.

collections, modifying training procedures, and designing regulatory incentives to encourage this alignment approach. We provide further concluding remarks in Appendix B.

## Acknowledgments

The authors gratefully acknowledge support from NSF grant CCF-2217062, the NSF Graduate Research Fellowship (grant DGE-2139899), a grant from the Simons Foundation, and a grant from the UK AI Safety Institute (AISI). The authors used AI tools (GPT-5 and Gemini 2.5 Pro within the Windsurf and Cursor environments) as an aid in writing code and proving lemmas, with careful instructions from the authors. All LLM-produced content was reviewed and edited by the authors before usage.

## Impact Statement

This paper studies AI alignment through the lens of *competition*: when a user can choose among multiple differently misaligned AI agents, strategic incentives can induce outcomes comparable to interacting with a perfectly aligned model. Our primary contribution is conceptual: we give a multi-leader Stackelberg framework for competitive, multi-round interactions and identify an explicit *market alignment* condition (approximate convex-hull containment of the user's utility by the agents' utilities) under which competitive equilibria provide strong welfare guarantees. If borne out in practice, these results suggest a complementary route to alignment that does not require any single model to be well aligned in isolation, but instead leverages diversity and competition.

The work also highlights potential risks and limitations. First, competitive dynamics are not universally beneficial: when the market alignment condition fails, our simulations indicate that competition may not reliably improve user outcomes, so the framework cautions against assuming that adding more models automatically yields safer or more helpful behavior. Second, the approach could be misapplied as a justification for deploying many strategically motivated systems without adequate oversight; in settings where agents share incentives, collude, or are controlled by a small number of actors, competition may not produce the intended benefits. Third, our model abstracts away from important real-world considerations such as unequal access, power asymmetries, manipulation, privacy concerns, and harms not captured by a single utility function. These limitations point to the need for careful auditing of model collections (e.g., testing approximate convex-hull coverage), transparency about incentives, and empirical validation before drawing policy conclusions. Overall, we view this work as an early step toward mechanism-design principles for alignment, complementing (rather than replacing) technical alignment

and governance safeguards.

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

# A. Additional Related Work

**Bayesian Persuasion with Competing Senders**   (Gradwohl et al., 2022) study a Bayesian persuasion game in which a receiver chooses to interact with only one of several competing senders (similar to our model in Section H). As we do, they find that competition can force senders to be fully informative in equilibrium. In addition to the greater generality of our setup beyond Bayesian persuasion, our work differs in its core assumptions. The assumption driving the results of (Gradwohl et al., 2022) is that the senders are uncertain about each other's utility functions, and that any sender has a non-zero probability of being perfectly aligned with the receiver. We instead introduce and use the arguably more general "approximate market alignment" assumption, which only requires the user's utility to lie within the convex hull of the AI agents' utilities — we do not require any uncertainty about the AI agent utility functions, or any possibility that any of them are individually aligned with the user. (Hossain et al., 2024) study the problem of multi-sender Bayesian Persuasion from a computational perspective, and prove worst-case hardness results for both the receiver's best-response problem and for the senders' equilibrium computation problem. They also design and evaluate neural network architectures suited to the (heuristic) computation of equilibria in such games.

**AI Debate**   Our work bears some similarity to AI alignment via "debate" as proposed by (Irving et al., 2018). In their setup, two AI agents take turns making arguments about some proposition (e.g. the factuality of some claim), and at the end one of them is chosen as the "winner" of the debate by a human user. The goal of each agent is only to be declared the winner, and so this is a two-player zero sum game. The hope is that the equilibrium strategy will be to be honest, because "it is harder to lie than to refute a lie." Several subsequent theoretical works have been motivated by AI safety via debate. For example, (Brown-Cohen et al., 2024; 2025) study multi-prover proof systems and study what kinds of problems have solutions such that an "honest prover" has a winning strategy implementable by a Turing machine of bounded complexity and a verifier that makes a bounded number of oracle calls to human judgment. (Chen et al., 2024) use AI debate as motivation for studying the problem of learning in very large zero sum games through use of an oracle. A main conceptual difference between our model and this literature is that we do not assume that the AI agents are motivated to be "chosen" as winners, but rather that they aim to influence Alice's behavior (in a complex decision space with non-binary actions and outcomes). Our work can be viewed as an extension of the AI debate model beyond two player zero sum games, to many LLMs who may have goals in common, but who desire to influence user behavior in different ways.

# B. Discussion and Conclusion

We have introduced a new approach to AI alignment—through competition between multiple, differently misaligned models so that the benefits of perfect alignment emerge in equilibrium. The key condition we need is *market alignment*, that the human user's utility function can be approximately represented as a non-negative weighted combination of the AI models' utility functions —i.e., up to scaling, that Alice's utility function lies approximately in the convex hull of the Bobs'. This is a much more robust (and easier to satisfy) condition than requiring any single AI model to be close to perfectly aligned.

We view our work as the first step in a broader research agenda of mechanism design for AI alignment. Our analysis is highly stylized; our paper assumes that the AI models are acting in equilibrium of a highly complex game (which are computationally hard to find even in simpler settings (Hossain et al., 2024)). It also assumes that Alice is able to use deployed AI models optimally, and act optimally given the information she learns from them —in particular, that Alice is able to correctly form posterior beliefs given the information she learns. We see two key future directions to our research agenda here: 1) developing protocols with more robust guarantees that do not depend on computationally intensive behavior on the part of the participants, and 2) designing platforms that aid participants in performing these cognitive tasks, for example by aggregating the user's past experiences with multiple AI models.

We have also studied a setting in which the strategic agents must *commit* to conversation rules (in the style of Bayesian Persuasion (Kamenica & Gentzkow, 2011)) — this is well motivated by current AI technology, in which models are represented by static weights which must be trained at significant expense before deployment and then represent conversation rules that users can interact with, without them maintaining significant state between sessions. However as AI agents become more stateful and dynamic across time, strategic models that do not involve commitment (and require that both parties are simultaneously best responding to one another's conversation rules, in the style of *cheap talk* (Crawford & Sobel, 1982; Farrell & Rabin, 1996)) may become more relevant. We expect that the tools of game theory and mechanism design will become important to understand the alignment of *marketplaces* of AI agents.

We have also modeled a *single* downstream user Alice. Alice could of course stand in for many users, but central to our

modeling is that Alice—and by extension, all of the users whom she is the stand-in for—have a single utility function. Of course, AI users do not actually have a single, monolithic utility function—this is the central concern of *pluralistic alignment* (Gabriel, 2020; Sorensen et al., 2024; Shirali et al., 2025). A natural extension of our work would consider a diverse population of downstream users. Since different users with different utility functions can best-respond to a fixed set of conversation rules differently, a tantalizing opportunity within such a model is that in equilibrium, a single set of fixed conversation rules might simultaneously give many downstream users the benefits of interacting with a fully aligned model, despite the fact that "fully aligned" means something different for each user.

Our model also suggests a number of ancillary questions. If our goal is to maintain marketplaces of models that approximately satisfy the market alignment condition for as many users as possible, how can we test or audit whether existing collections of models do? How can we modify training procedures to optimize for this condition? What kinds of regulatory and economic incentives would encourage AI model providers to aim for it?

## C. Additional Preliminaries

Here we give a formal procedure for sampling a transcript given by the game defined in Section 2.

---

**Algorithm 1** SAMPLETRANSCRIPT$(\vec{C_B}, C_A)$: A protocol for sampling a transcript.

---

**Input:** Conversation rules $\vec{C_B}, C_A$.
**Output:** A transcript $\pi = (m_1, \ldots, m_k)$ where $m_i$ is the history of messages between Alice and Bob$_i$.
  Initialize empty histories $h_i = ()$ for all $i \in [k]$.
  **for** $r = 1, \ldots, R$ **do**
    Alice sends a message to each Bob: $(m_{A,1}, \ldots, m_{A,k}) \sim C_A(x_A, (h_1, \ldots, h_k))$.
    Append messages to histories: $h_i \leftarrow h_i \circ m_{A,i}$ for all $i$.
    **for** each $i \in [k]$ **do**
      Bob $i$ sends a message to Alice: $m_{B,i} \sim C_{B_i}(x_B, h_i)$.
      Append message to history: $h_i \leftarrow h_i \circ m_{B,i}$.
  **return** transcript $\pi = (h_1, \ldots, h_k)$.

---

### C.1. Equilibrium Existence

Nash equilibria exist in our setting under the same conditions under the same conditions under which they are known to exist in multi-sender Bayesian Persuasion (Gentzkow & Kamenica, 2017a;b; Hossain et al., 2024). (Gentzkow & Kamenica, 2017a;b) show constructively that the set of Nash equilibria is non-empty by constructing a *fully disclosive* equilibrium whenever full disclosure is in the strategy space of the senders. In our setting that corresponds to the existence of a conversation rule $C$ such that for all $x_A, x_B$, Alice's best response decision rule $D_A^*(C)$ places all of its weight on $a \in \arg\max_{a \in \mathcal{A}} \mathbb{E}_y[u_A(a, y)|x_A, x_B]$ (a condition we also use in our simplest results in Section 3, Proposition 3.2). If there are at least two Bobs who are both playing this conversation rule, then neither one of them can affect Alice's induced outcome distribution through a unilateral deviation, and hence this is an equilibrium (establishing that the set of Nash equilibria is non-empty). (Hossain et al., 2024) extend this line of argument to settings in which full disclosure is not in the strategy space of any sender individually, but there is a coalition of senders that can yield full disclosure in a way that is robust to any single deviation; they show that this is possible (via an error correcting code construction) whenever the message space is sufficiently large. All of our theorems characterize the full set of Nash equilibria in our setting, and so apply non-trivially in any setting in which the set of Nash equilibria is non-empty.

## D. A Probabilistic Motivation for Approximate Market Alignment

The approximate market alignment assumption (Definition 2.1) is central to our results, but where might it come from? Here, we provide a simple generative model for AI agent utilities under which the assumption holds with high probability for a sufficiently large set of agents. This models the scenario described in the introduction where AI agents are designed to be aligned with the human user (i.e., aligned in expectation) but their implementation is imperfect due to the difficulty of the alignment problem.

**A Random Utility Model.** Suppose each AI agent's utility function is drawn independently from a distribution. We assume that for any action $a \in \mathcal{A}$ and any state of the world $y \in \mathcal{Y}$, the expected utility of any AI agent $i$ is equal to Alice's utility. That is,

$$\mathbb{E}[U_i(a, y)] = u_A(a, y).$$

We also assume all utilities are bounded, $U_i(a, y) \in [0, 1]$.

This model captures the intuition from our introduction: if each AI developer attempts to create an aligned model but fails due to implementation noise, then the simple average of many such models will be well-aligned. This is the "concentration of measure" effect mentioned in the introduction—while any individual model may be poorly aligned, the average converges to the target.

Under this model, we can show that a sufficiently large set of AI agents will satisfy the $\varepsilon$-market alignment condition with uniform weights ($w_i = 1/k$) and zero offset ($c = 0$). This follows from a standard concentration inequality argument.

**Proposition D.1** (Market Alignment from Noisy Implementation). *Let the utility functions for a set of $k$ AI agents be drawn independently according to the random utility model above. Assume the action space $\mathcal{A}$ and state space $\mathcal{Y}$ are finite. Then for any alignment tolerance $\varepsilon > 0$ and any failure probability $\delta > 0$, if the number of agents $k$ satisfies*

$$k > \frac{\ln(2|\mathcal{A}||\mathcal{Y}|) - \ln(\delta)}{2\varepsilon^2},$$

*then the agents satisfy the $\varepsilon$-market alignment condition (Definition 2.1) with uniform weights $w_i = 1/k$ and zero offset $c = 0$, with probability at least $1 - \delta$.*

*Proof.* For any fixed action-state pair $(a, y)$, the random variables $U_1(a, y), \ldots, U_k(a, y)$ are independent and bounded in $[0, 1]$. Let $\bar{U}(a, y) = \frac{1}{k} \sum_{i=1}^{k} U_i(a, y)$ be their sample mean. By Hoeffding's inequality, the probability of a large deviation from the true mean $u_A(a, y)$ is bounded:

$$\mathbb{P}(|\bar{U}(a, y) - u_A(a, y)| > \varepsilon) \leq 2e^{-2k\varepsilon^2}.$$

For the approximate average alignment condition to fail, this deviation must occur for at least one pair $(a, y) \in \mathcal{A} \times \mathcal{Y}$. We can bound the probability of this event using a union bound over all possible pairs:

$$\mathbb{P}(\sup_{a,y} |\bar{U}(a, y) - u_A(a, y)| > \varepsilon) = \mathbb{P}(\exists (a, y) \in \mathcal{A} \times \mathcal{Y} \text{ s.t. } |\bar{U}(a, y) - u_A(a, y)| > \varepsilon)$$

$$\leq \sum_{(a,y) \in \mathcal{A} \times \mathcal{Y}} \mathbb{P}(|\bar{U}(a, y) - u_A(a, y)| > \varepsilon)$$

$$\leq |\mathcal{A}||\mathcal{Y}| \cdot 2e^{-2k\varepsilon^2}.$$

We want this failure probability to be less than $\delta$. So, we set

$$|\mathcal{A}||\mathcal{Y}| \cdot 2e^{-2k\varepsilon^2} < \delta.$$

Solving for $k$, we take the logarithm of both sides:

$$\ln(2|\mathcal{A}||\mathcal{Y}|) - 2k\varepsilon^2 < \ln(\delta)$$

$$-2k\varepsilon^2 < \ln(\delta) - \ln(2|\mathcal{A}||\mathcal{Y}|)$$

$$2k\varepsilon^2 > \ln(2|\mathcal{A}||\mathcal{Y}|) - \ln(\delta)$$

$$k > \frac{\ln(2|\mathcal{A}||\mathcal{Y}|) - \ln(\delta)}{2\varepsilon^2}.$$

Thus, if $k$ meets this condition, the probability that the set of AI agents does not satisfy our $\varepsilon$-market alignment assumption is less than $\delta$. The probability of successful alignment is therefore at least $1 - \delta$. $\square$

This result shows that the number of AI agents required grows logarithmically with the size of the action and state spaces, and polynomially with respect to $1/\varepsilon$. It provides a clear and direct path to satisfying our key assumption by simply having a large enough population of imperfectly-aligned agents.

# E. Proof of Proposition 3.2

*Proof.* Suppose a leader $i \in S$ unilaterally deviates to the Alice-optimal conversation rule $C_B^*$. By assumption, Alice has a conversation rule that would allow her to learn her Bayes-optimal action, $a^*(x_A, x_B)$ by interacting with $C_B^*$. Alice's strategy space includes the option of ignoring all Bobs other than $i$ and playing her best response as if it were a single-leader game with leader $i$. Since Alice plays a best-response to the full set of strategies $\vec{C_B}$, her utility must be at least as high as what she could get from this simpler strategy.

When Alice learns the specific action $a^*(x_A, x_B)$, her best response is to play that action (or a distribution over optimal actions if there are ties, according to her fixed tie-breaking rule). This response depends only on the information she learns, not on the identity of the Bob who provided it, since we assume that ties amongst her best response actions are broken according to a fixed tie breaking rule. Therefore, if any Bob $j \in S$ deviates to $C_B^*$, Alice will follow the same decision rule.

Consequently, the induced distribution over actions and outcomes, $\mathcal{I}^*((\vec{C_B}^{-i}, C_B^*))$, is identical for any deviating Bob $i \in S$. Thus, the condition is satisfied. $\square$

# F. Proof of Theorem 3.4

*Proof.* Fix an arbitrary Nash equilibrium $\vec{C_B}$ and let $\mathcal{I}_{NE} = \mathcal{I}^*(\vec{C_B})$ be the distribution induced by the equilibrium strategies. Now, consider a unilateral deviation by an arbitrary Bob $i$ to the Alice-optimal strategy $C_B^*$. Let $\mathcal{I}_{dev} = \mathcal{I}^*((\vec{C_B}^{-i}, C_B^*))$ be the induced distribution after this deviation. By the identical induced distribution condition, $\mathcal{I}_{dev}$ is the same regardless of which Bob $i$ deviates.

When a single Bob $i$ deviates to using the conversation rule $C_B^*$, Alice's strategy space includes the option of ignoring all other Bob's $j$ and engaging with Bob $i$ as she would in the single-leader game. Since Alice chooses a best-response strategy, her resulting utility must be at least as high as the utility from this option, which is by definition $U_A(C_B^*)$.

$$\mathbb{E}_{\mathcal{I}_{dev}}[u_A(a, y)] \geq U_A(C_B^*).$$

By the Nash equilibrium condition, no Bob $i$ has an incentive to deviate. Thus, for all $i \in [k]$:

$$\mathbb{E}_{\mathcal{I}_{dev}}[U_i(a, y)] \leq \mathbb{E}_{\mathcal{I}_{NE}}[U_i(a, y)].$$

Taking a weighted sum over all Bobs using the non-negative weights $w_i$ from the alignment assumption (where $\sum w_i = 1$):

$$\sum_{i=1}^{k} w_i \mathbb{E}_{\mathcal{I}_{dev}}[U_i(a, y)] \leq \sum_{i=1}^{k} w_i \mathbb{E}_{\mathcal{I}_{NE}}[U_i(a, y)].$$

By linearity of expectation, this is equivalent to:

$$\mathbb{E}_{\mathcal{I}_{dev}}\left[\sum_{i=1}^{k} w_i U_i(a, y)\right] \leq \mathbb{E}_{\mathcal{I}_{NE}}\left[\sum_{i=1}^{k} w_i U_i(a, y)\right].$$

Now we use the approximate market alignment assumption, which states that $\sum w_i U_i(a, y)$ is $\varepsilon$-close to $u_A(a, y) - c$. For the left-hand side:

$$\mathbb{E}_{\mathcal{I}_{dev}}\left[\sum_{i=1}^{k} w_i U_i(a, y)\right] \geq \mathbb{E}_{\mathcal{I}_{dev}}[u_A(a, y) - c] - \varepsilon = \mathbb{E}_{\mathcal{I}_{dev}}[u_A(a, y)] - c - \varepsilon \geq U_A(C_B^*) - c - \varepsilon.$$

For the right-hand side:

$$\mathbb{E}_{\mathcal{I}_{NE}}\left[\sum_{i=1}^{k} w_i U_i(a, y)\right] \leq \mathbb{E}_{\mathcal{I}_{NE}}[u_A(a, y) - c] + \varepsilon = \mathbb{E}_{\mathcal{I}_{NE}}[u_A(a, y)] - c + \varepsilon.$$

Combining these inequalities, we get:

$$U_A(C_B^*) - c - \varepsilon \leq \mathbb{E}_{\mathcal{I}_{NE}}[u_A(a, y)] - c + \varepsilon.$$

The constant offset $c$ cancels, and we are left with:

$$U_A(C_B^*) - \varepsilon \leq \mathbb{E}_{\mathcal{I}_{NE}}[u_A(a, y)] + \varepsilon.$$

$$\mathbb{E}_{\mathcal{I}_{NE}}[u_A(a, y)] \geq U_A(C_B^*) - 2\varepsilon.$$

which completes the proof. $\qquad\square$

## G. Robust Guarantees for Users with Bounded Rationality

The result in Section 3 required that the *identical induced distribution condition* (Definition 3.1) was satisfied. This condition is always satisfied when a perfectly aligned model could cause Alice to learn her exact Bayes optimal action. However, this is a strong assumption. Here we relax our motivating assumption to a more realistic condition: a perfectly aligned model need only help Alice approximately learn the expected utility of each action. We show that this implies a relaxation of our main technical condition — an approximate version of the identical induced distribution condition (Definition G.6) which we use in this section.

To analyze this weaker setting, we study a model where Alice acts straightforwardly rather than strategically, committing to two behavioral rules: (1) she always reports her honest beliefs during conversation, and (2) she uses "quantal response" for decision-making—a form of bounded rationality where she chooses actions probabilistically based on their estimated utilities rather than always picking the best one. We can view these assumptions either as modeling a boundedly rational Alice, or as explicit behavioral commitments that a strategic Alice makes to enjoy more robust guarantees.

This section proceeds in three steps. First, we introduce the quantal response model where Alice commits to straightforward conversation and bounded rational decision-making (Appendix G.1). Second, we prove that this leads to near-optimal utility in equilibrium under a technical condition relaxing the identical induced distribution condition (Appendix G.2). Finally, we show this condition is satisfied when the underlying distribution has the "information substitutes" property (Appendix G.3). The main result (Theorem G.15) shows that under the Information Substitutes condition, Alice achieves near-optimal utility with an explicit bound depending on alignment error, estimation error, and the quantal response gap.

### G.1. The Quantal Response Model

In this model, we assume Alice reacts to any set of conversation rules that the Bobs commit to using a straightforward conversation rule and a quantal response decision rule. This can be viewed either as a model of nonstrategic interaction and bounded rationality or as a strategic commitment by Alice to encourage more informative communication.

**Definition G.1** (Straightforward Conversation Rule). *The straightforward conversation rule models honest communication: at each round, a player simply reports their current beliefs about the expected utility of each action. This can be viewed either as modeling non-strategic behavior or as a commitment device to encourage informative equilibria.*

*Specifically, let $\pi_i^{k-1}$ denote the private transcript between Alice and Bob $i$ up to round $k-1$, and let $\vec{\pi}^{k-1} = (\pi_1^{k-1}, \ldots, \pi_k^{k-1})$ be the full history available to Alice. If Alice uses the straightforward conversation rule, her message is $m_A^k = (\mathbb{E}[u_A(a, y) \mid x_A, \vec{\pi}^{k-1}])_{a \in \mathcal{A}}$. If Bob $i$ uses the straightforward conversation rule, his message is $m_{B_i}^k = (\mathbb{E}[u_A(a, y) \mid x_{B_i}, \pi_i^{k-1}])_{a \in \mathcal{A}}$. We assume the message space $\mathcal{M}$ is sufficiently expressive to encode these vectors, e.g., $[0, 1]^{|\mathcal{A}|} \subseteq \mathcal{M}$. We denote Alice's use of this rule as $C_A^{sf}$.*

We model Alice as choosing her action using quantal response, a model of bounded rationality from behavioral economics (McKelvey & Palfrey, 1995).

**Definition G.2** (Quantal Response Decision Rule). *Rather than always choosing the action with highest estimated utility (which would be "best response"), Alice uses quantal response: she chooses actions probabilistically, with higher-utility actions being more likely. The parameter $\lambda$ controls how "rational" she is—as $\lambda \to \infty$, this approaches best response.*

*Formally, given Alice's features $x_A$ and the final transcript $\vec{\pi}$, from which she forms the posterior belief $\mu(x_A, \vec{\pi}) = (\mu_a(x_A, \vec{\pi}))_{a \in \mathcal{A}}$, the probability of choosing action $a$ is:*

$$D_A^Q(x_A, \vec{\pi})(a) = \frac{\exp\left(\lambda \mu_a(x_A, \vec{\pi})\right)}{\sum_{a' \in \mathcal{A}} \exp\left(\lambda \mu_{a'}(x_A, \vec{\pi})\right)}.$$

In this version of the game, Alice commits to both a fixed conversation rule, $C_A^{sf}$, and a fixed decision rule, $D_A^Q$. The Bobs, knowing this, choose their conversation rules to form a Nash Equilibrium.

**Definition G.3** (Quantal Response Equilibrium). *Let Alice's conversation rule $C_A$ be fixed to the straightforward conversation rule $C_A^{sf}$ and her decision rule be fixed to the $\lambda$-quantal rule $D_A^Q$.*

*Let $\mathcal{I}^Q(\vec{C}_B) = \mathcal{I}(\vec{C}_B, C_A^{sf}, D_A^Q)$ be the induced distribution given a vector of Bob strategies $\vec{C}_B$. A strategy profile $\vec{C}_B^{\,*}$ is a Quantal Response Nash Equilibrium if for all Bobs $i$ and all alternative rules $C'_{B_i}$:*

$$\mathbb{E}_{(a,y)\sim\mathcal{I}^Q(\vec{C}_B^{\,*})}[U_i(a,y)] \geq \mathbb{E}_{(a,y)\sim\mathcal{I}^Q((C'_{B_i},\vec{C}_{B,-i}^*))}[U_i(a,y)].$$

For reasonable values of $\lambda$, the quantal response decision rule gives Alice nearly as much utility as the best response decision rule, in expectation. As $\lambda$ grows large quantal response approaches best response. The next lemma formalizes this.

**Lemma G.4** (Quantal Response Gap). *For any belief vector $\mu$, the gap between the optimal utility and the expected utility from a $\lambda$-quantal response is bounded:*

$$\max_{a'\in\mathcal{A}}\mu_{a'} - \sum_{a\in\mathcal{A}}\frac{\exp(\lambda\mu_a)}{\sum_{a''\in\mathcal{A}}\exp(\lambda\mu_{a''})}\mu_a \leq \frac{\log|\mathcal{A}|}{\lambda}.$$

*Proof.* Let $a^* = \arg\max_{a\in\mathcal{A}}\mu_a$ be an optimal action and let $p(a) = \frac{\exp(\lambda\mu_a)}{\sum_{a'\in\mathcal{A}}\exp(\lambda\mu_{a'})}$ be the probability of choosing action $a$ under the quantal response model, for brevity. The optimal utility given belief $\mu$ is $\mu_{a^*}$. The expected utility under quantal response is $\sum_{a\in\mathcal{A}}p(a)\mu_a$.

The difference is:

$$\mu_{a^*} - \sum_{a\in\mathcal{A}}p(a)\mu_a = \sum_{a\in\mathcal{A}}p(a)(\mu_{a^*} - \mu_a).$$

From the definition of $p(a)$, we have $\mu_a = \frac{1}{\lambda}\log(p(a)Z)$, where $Z = \sum_{a'}\exp(\lambda\mu_{a'})$. Substituting this in:

$$\mu_{a^*} - \sum_{a\in\mathcal{A}}p(a)\mu_a = \sum_{a\in\mathcal{A}}p(a)\left(\mu_{a^*} - \frac{1}{\lambda}(\log p(a) + \log Z)\right)$$

$$= \mu_{a^*} - \frac{1}{\lambda}\left(\sum_{a\in\mathcal{A}}p(a)\log p(a) + \log Z\sum_{a\in\mathcal{A}}p(a)\right)$$

$$= \mu_{a^*} + \frac{H(p)}{\lambda} - \frac{\log Z}{\lambda},$$

where $H(p)$ is the Shannon entropy of the distribution $p$. Since $Z = \sum_{a'}\exp(\lambda\mu_{a'}) \geq \exp(\lambda\mu_{a^*})$, we have $\log Z \geq \lambda\mu_{a^*}$. Therefore,

$$\mu_{a^*} - \sum_{a\in\mathcal{A}}p(a)\mu_a \leq \mu_{a^*} + \frac{H(p)}{\lambda} - \frac{\lambda\mu_{a^*}}{\lambda} = \frac{H(p)}{\lambda}.$$

The entropy $H(p)$ is maximized when $p$ is the uniform distribution over $\mathcal{A}$, in which case $H(p) = \log|\mathcal{A}|$. Thus, we have the bound:

$$\max_{a'\in\mathcal{A}}\mu_{a'} - \sum_{a\in\mathcal{A}}D_A^Q(\pi)(a)\mu_a \leq \frac{\log|\mathcal{A}|}{\lambda}.$$

$\square$

**Lemma G.5** (Multiplicative Stability of Quantal Response). *Let $P = \mathrm{softmax}(\lambda u)$ and $Q = \mathrm{softmax}(\lambda u')$ over $\mathcal{A}$, where for a vector $z \in \mathbb{R}^{\mathcal{A}}$ we define $\mathrm{softmax}(z)_a := \exp(z_a)/\sum_{b\in\mathcal{A}}\exp(z_b)$. If $\|u - u'\|_\infty \leq \varepsilon$, then for each $a \in \mathcal{A}$,*

$$e^{-2\lambda\varepsilon} \leq \frac{P(a)}{Q(a)} \leq e^{2\lambda\varepsilon}.$$

*Consequently,*

$$\|P - Q\|_1 \leq e^{2\lambda\varepsilon} - 1.$$

*Proof.* For any $a$, $e^{-\lambda\varepsilon} \leq \frac{e^{\lambda u_a}}{e^{\lambda u'_a}} \leq e^{\lambda\varepsilon}$, and for the partition functions $Z = \sum_b e^{\lambda u_b}$, $Z' = \sum_b e^{\lambda u'_b}$ we have $e^{-\lambda\varepsilon} Z' \leq Z \leq e^{\lambda\varepsilon} Z'$. Therefore $e^{-2\lambda\varepsilon} \leq \frac{P(a)}{Q(a)} = \frac{e^{\lambda u_a}/Z}{e^{\lambda u'_a}/Z'} \leq e^{2\lambda\varepsilon}$. Then $|P(a) - Q(a)| = Q(a)\,|P(a)/Q(a) - 1| \leq Q(a)(e^{2\lambda\varepsilon} - 1)$. Summing over $a$ gives $\|P - Q\|_1 \leq e^{2\lambda\varepsilon} - 1$. $\qquad\square$

### G.2. Equilibrium Analysis Under the $(\delta, C_B^*)$-Close Condition

Our goal in this subsection is to prove a bound on Alice's utility in any equilibrium of the induced game (Theorem G.7). Our proof strategy has several parts. First, to reason about equilibria, we need a way to compare the outcomes that result from different Bobs' strategies. We formalize this with the *$(\delta, C_B^*)$-close condition* (Theorem G.6), which states that any two Bobs unilaterally deviating to a reference strategy $C_B^*$ should induce similar outcome distributions. Second, we show that this condition holds if the reference strategy allows Alice to learn her expected utility for each action with small error (Theorem G.8). In Appendix G.3, we will show how the Information Substitutes condition provides a foundation for bounding this error, completing our argument.

**Definition G.6** (($\delta, C_B^*$)-Close Condition). *This condition captures the idea that it "doesn't matter" which Bob adopts the reference strategy $C_B^*$—the resulting outcomes are similar regardless. Intuitively, this holds when $C_B^*$ allows Alice to learn something fundamental about the world state, rather than Bob-specific information.*

*Formally, we say that a game satisfies the $(\delta, C_B^*)$-**close condition** for a reference strategy $C_B^*$ if for any strategy profile $\vec{C}_B$ and any two Bobs $i, j$, the total variation distance between the induced distributions resulting from their unilateral deviations to $C_B^*$ is at most $\delta$:*

$$\|\mathcal{I}^Q((\vec{C}_{B,-i}, C_B^*)) - \mathcal{I}^Q((\vec{C}_{B,-j}, C_B^*))\|_1 \leq \delta.$$

We first prove a general result: if any Bob unilaterally adopting a reference strategy $C_B^*$ would induce approximately the same outcome distribution (the $(\delta, C_B^*)$-close condition), then Alice's utility in any equilibrium of the induced game is close to the utility she would get from interacting with that single Bob using conversation rule $C_B^*$.

**Theorem G.7** (Equilibrium Utility Bound with Quantal Response). *Suppose the leaders Bob satisfy the $\varepsilon$-market alignment condition and the game satisfies the $(\delta, C_B^*)$-close condition (Theorem G.6) for a reference strategy $C_B^*$. Let $U_A(C_B^*)$ be Alice's expected utility from interacting with a single Bob using $C_B^*$. Then in any Quantal Response Nash Equilibrium (Theorem G.3), her expected utility is at least:*

$$\mathbb{E}_{\mathcal{I}_{NE}^Q}[u_A] \geq U_A(C_B^*) - 2\varepsilon - \delta.$$

*Proof.* Fix a Quantal Response Nash Equilibrium $\vec{C}_B^*$ with induced distribution $\mathcal{I}_{NE}^Q = \mathcal{I}^Q(\vec{C}_B^*)$. Let $\mathcal{I}_{dev,j} = \mathcal{I}^Q((\vec{C}_{B,-j}^*, C_B^*))$ be the distribution induced when Bob $j$ unilaterally deviates. By the definition of a Quantal Response Nash Equilibrium, no Bob $j \in [k]$ has an incentive to deviate. This implies that for every $j \in [k]$:

$$\mathbb{E}_{\mathcal{I}_{dev,j}}[U_j] \leq \mathbb{E}_{\mathcal{I}_{NE}^Q}[U_j].$$

Taking a weighted sum with weights $w_j \geq 0$ where $\sum w_j = 1$:

$$\sum_{j=1}^{k} w_j \mathbb{E}_{\mathcal{I}_{dev,j}}[U_j] \leq \sum_{j=1}^{k} w_j \mathbb{E}_{\mathcal{I}_{NE}^Q}[U_j].$$

The right-hand side can be bounded using the market alignment assumption:

$$\sum_{j=1}^{k} w_j \mathbb{E}_{\mathcal{I}_{NE}^Q}[U_j] = \mathbb{E}_{\mathcal{I}_{NE}^Q}\left[\sum_j w_j U_j\right] \leq \mathbb{E}_{\mathcal{I}_{NE}^Q}[u_A - c] + \varepsilon = \mathbb{E}_{\mathcal{I}_{NE}^Q}[u_A] - c + \varepsilon.$$

For the left-hand side, we first relate each term to a single anchor deviation by an arbitrary leader Bob $k$. Let $\mathcal{I}_{dev,k}$ be the distribution induced by Bob $k$'s deviation. The utility for Bob $j$ under their own deviation $\mathcal{I}_{dev,j}$ is close to their utility under the anchor deviation $\mathcal{I}_{dev,k}$:

$$|\mathbb{E}_{\mathcal{I}_{dev,j}}[U_j] - \mathbb{E}_{\mathcal{I}_{dev,k}}[U_j]| \leq \|\mathcal{I}_{dev,j} - \mathcal{I}_{dev,k}\|_1 \leq \delta.$$

Therefore, $\mathbb{E}_{\mathcal{I}_{dev,j}}[U_j] \geq \mathbb{E}_{\mathcal{I}_{dev,k}}[U_j] - \delta$. Applying this to the weighted sum:

$$\sum_{j=1}^{k} w_j \mathbb{E}_{\mathcal{I}_{dev,j}}[U_j] \geq \sum_{j=1}^{k} w_j (\mathbb{E}_{\mathcal{I}_{dev,k}}[U_j] - \delta) = \left(\sum_{j=1}^{k} w_j \mathbb{E}_{\mathcal{I}_{dev,k}}[U_j]\right) - \delta \sum_{j} w_j.$$

Since $\sum w_j = 1$, this simplifies to $\mathbb{E}_{\mathcal{I}_{dev,k}}[\sum_j w_j U_j] - \delta$. We now apply the alignment assumption to this term:

$$\mathbb{E}_{\mathcal{I}_{dev,k}}\left[\sum_j w_j U_j\right] - \delta \geq (\mathbb{E}_{\mathcal{I}_{dev,k}}[u_A] - c - \varepsilon) - \delta.$$

By definition, Alice's utility from this single-Bob deviation is $\mathbb{E}_{\mathcal{I}_{dev,k}}[u_A] = U_A(C_B^*)$. So the LHS is bounded below by $U_A(C_B^*) - c - \varepsilon - \delta$.

Putting the full inequality back together:

$$U_A(C_B^*) - c - \varepsilon - \delta \leq \mathbb{E}_{\mathcal{I}_{NE}^Q}[u_A] - c + \varepsilon.$$

The constant offset $c$ cancels, and rearranging yields theorem:

$$U_A(C_B^*) - 2\varepsilon - \delta \leq \mathbb{E}_{\mathcal{I}_{NE}^Q}[u_A].$$

$\square$

Next we show that any conversation rule that in the single leader game would cause Alice to approximately learn the utility of each of her actions satisfies the approximate closeness condition needed to invoke Theorem G.7.

**Proposition G.8** (Uniform Utility Estimation Error Implies $\delta$-Close). *Suppose a reference conversation rule $C_B^*$ is such that when used with Alice's fixed straightforward conversation rule $C_A^{sf}$, the utility estimates are uniformly accurate across actions, almost surely: for all $(x_A, x_B)$ and all transcripts $\vec{\pi}$ generated under $(C_A^{sf}, C_B^*)$, we have $\|\mu(x_A, \vec{\pi}) - \mu_{true}(x_A, x_B)\|_\infty \leq \varepsilon_u$. Then the game satisfies the $(\delta, C_B^*)$-close condition (Theorem G.6) with $\delta \leq e^{4\lambda\varepsilon_u} - 1$.*

*Proof.* Let $\vec{C}_B$ be an arbitrary vector of Bobs' strategies. The distribution $\mathcal{I}_{dev,i}$ is induced when Bob $i$ unilaterally deviates to a reference strategy $C_B^*$, so the vector of Bobs' strategies is $(\vec{C}_{B,-i}, C_B^*)$. Similarly, for Bob $j$'s deviation, the strategy vector is $(\vec{C}_{B,-j}, C_B^*)$. Our goal is to show that the total variation distance between the induced distributions $\mathcal{I}^Q((\vec{C}_{B,-i}, C_B^*))$ and $\mathcal{I}^Q((\vec{C}_{B,-j}, C_B^*))$ is bounded, for any $\vec{C}_B$.

The induced distributions from the deviations by $i$ and $j$ are joint distributions over Alice's action $a$ and the world state $y$. Let $P_i(a, y)$ and $P_j(a, y)$ denote these distributions. We first show that the total variation distance between them is bounded by the expected distance between Alice's action distributions, conditioned on the features.

By the law of total probability, the joint distribution $P_k(a, y)$ (for $k \in \{i, j\}$) is given by integrating over the features $(x_A, x_B)$:

$$P_k(a, y) = \int_{\mathcal{X}_A, \mathcal{X}_B} P(x_A, x_B) P_k(a, y \mid x_A, x_B) \, dx_A dx_B = \mathbb{E}_{(x_A, x_B)}[P_k(a, y \mid x_A, x_B)].$$

The conditional distribution $P_k(a, y \mid x_A, x_B)$ factors according to the causal structure of the game: first a transcript $\pi$ is generated, then an action $a$ is chosen. The state $y$ is conditionally independent of the transcript and action given the features. Thus, $P_k(a, y \mid x_A, x_B) = P(y \mid x_A, x_B) q_k(a \mid x_A, x_B)$, where $q_k(a \mid x_A, x_B)$ is Alice's action probability given the features under deviation $k$.

Now we bound the total variation distance:

$$\|\mathcal{I}_{dev,i} - \mathcal{I}_{dev,j}\|_1 = \sum_{a \in \mathcal{A}} \int_{\mathcal{Y}} |P_i(a,y) - P_j(a,y)|\, dy$$

$$= \sum_a \int_y |\mathbb{E}_{(x_A,x_B)}[P(y|x_A,x_B)(q_i(a|x_A,x_B) - q_j(a|x_A,x_B))]|\, dy$$

$$\leq \sum_a \int_y \mathbb{E}_{(x_A,x_B)}[P(y|x_A,x_B)|q_i(a|x_A,x_B) - q_j(a|x_A,x_B)|]\, dy \quad \text{(Jensen's Ineq.)}$$

$$= \mathbb{E}_{(x_A,x_B)}\left[ \sum_a |q_i(a|x_A,x_B) - q_j(a|x_A,x_B)| \int_y P(y|x_A,x_B)\, dy \right] \quad \text{(Fubini's Thm.)}$$

$$= \mathbb{E}_{(x_A,x_B)}\left[ \sum_a |q_i(a|x_A,x_B) - q_j(a|x_A,x_B)| \right] \quad \left(\text{since } \int_y P(y|\cdot)dy = 1\right)$$

$$= \mathbb{E}_{(x_A,x_B)}[\|q_i(\cdot|x_A,x_B) - q_j(\cdot|x_A,x_B)\|_1]$$

Here, $q_k(\cdot|x_A,x_B) = \mathbb{E}_{\vec{\pi} \sim \Pi_k(x_A,x_B)}[D_A^Q(x_A,\vec{\pi})]$ is Alice's action distribution for a given $(x_A,x_B)$, averaged over all possible transcripts $\vec{\pi}$ that could be generated when the Bobs' strategies are $(\vec{C}_{B,-k}, C_B^*)$.

Let $\mu(x_A,\vec{\pi})$ be Alice's posterior utility vector and let $\mu_{true}(x_A,x_B)$ be the true expected utility vector. Let $\vec{\pi}_i$ and $\vec{\pi}_j$ be random variables for the transcripts generated under deviations by $i$ and $j$ respectively. By the uniform-accuracy hypothesis, for all $(x_A,x_B)$ and all transcripts we have $\|\mu(x_A,\vec{\pi}_i) - \mu_{true}(x_A,x_B)\|_\infty \leq \varepsilon_u$ and $\|\mu(x_A,\vec{\pi}_j) - \mu_{true}(x_A,x_B)\|_\infty \leq \varepsilon_u$. Hence $\|\mu(x_A,\vec{\pi}_i) - \mu(x_A,\vec{\pi}_j)\|_\infty \leq 2\varepsilon_u$ deterministically.

Conditioning on $(x_A,x_B,\vec{\pi}_i,\vec{\pi}_j)$, apply Theorem G.5 with $\varepsilon = 2\varepsilon_u$ to the quantal response distributions to obtain

$$\|D_A^Q(x_A,\vec{\pi}_i) - D_A^Q(x_A,\vec{\pi}_j)\|_1 \leq e^{4\lambda\varepsilon_u} - 1.$$

Taking expectations over $(x_A,x_B,\vec{\pi}_i,\vec{\pi}_j)$ preserves the bound, and by the reduction above from joint to action-marginal differences we conclude

$$\|\mathcal{I}_{dev,i} - \mathcal{I}_{dev,j}\|_1 \leq e^{4\lambda\varepsilon_u} - 1.$$

This proves the claim with $\delta = e^{4\lambda\varepsilon_u} - 1$. $\qquad\qquad\qquad\qquad\qquad\qquad\qquad\qquad\qquad\qquad\square$

**Corollary G.9** (High-Probability Uniform Error Implies $\delta$-Close). *Under the setup of Theorem G.8, suppose there exists an event $E$ with probability at least $1 - \rho$ over $(x_A,x_B)$ and transcript randomness such that for all $(x_A,x_B) \in E$ and all transcripts $\vec{\pi}$ generated under $(C_A^{sf}, C_B^*)$, we have uniform accuracy across actions: $\|\mu(x_A,\vec{\pi}) - \mu_{true}(x_A,x_B)\|_\infty \leq \varepsilon_u$. Then the game satisfies the $(\delta, C_B^*)$-close condition with*

$$\delta \leq e^{4\lambda\varepsilon_u} - 1 + \rho.$$

*Proof.* On the event $E$, Theorem G.8 applies directly, yielding total variation at most $e^{4\lambda\varepsilon_u} - 1$. On the complement $E^c$ (probability at most $\rho$), total variation is at most 1. Taking expectations gives $\delta \leq (e^{4\lambda\varepsilon_u} - 1) \cdot (1 - \rho) + 1 \cdot \rho \leq e^{4\lambda\varepsilon_u} - 1 + \rho$. $\qquad\qquad\square$

**Corollary G.10** (Small-$\lambda$ Linearization). *If $\lambda\varepsilon_u \leq c$, then by the mean value theorem $e^{4\lambda\varepsilon_u} - 1 \leq 4\lambda\varepsilon_u\, e^{4c}$. In particular, if $\lambda\varepsilon_u \leq \frac{1}{4}$, then $\delta \leq 4e\, \lambda\varepsilon_u$.*

### G.3. From Information Substitutes to Utility Guarantees

The previous results provide a utility bound for Alice that depends on two key quantities: the utility estimation error $\varepsilon_u$ and the alignment error $\varepsilon$. We now show how the *Information Substitutes Condition* first defined in (Frongillo et al., 2021) (Theorem G.11) provides a foundation for bounding $\varepsilon_u$ when Alice uses the straightforward conversation rule, leading to our main theorem.

The Information Substitutes condition, roughly speaking, says that Alice's and Bob's information are "substitutes" rather than "complements" for predicting Alice's utility. If Alice already knows Bob's information, learning her own information

doesn't help as much, and vice versa. This is a reasonable assumption in many settings—for example, if both Alice and Bob observe noisy versions of the same underlying signal.

**Definition G.11** (Information Substitutes Condition (Frongillo et al., 2021)). *A distribution $P(x_A, x_B, y)$ satisfies the information substitutes condition with respect to Alice's utility function $u_A$ if, for every action $a \in \mathcal{A}$ and every pair of feature subsets $A \subseteq \mathcal{X}_A$ and $B \subseteq \mathcal{X}_B$, the following inequality holds:*

$$
\begin{aligned}
&\mathbb{E}\left[(u_A(a,y) - \mathbb{E}[u_A(a,y) \mid x_A \in A, x_B])^2 \mid x_A \in A, x_B \in B\right] \\
&\quad - \mathbb{E}\left[(u_A(a,y) - \mathbb{E}[u_A(a,y) \mid x_A, x_B])^2 \mid x_A \in A, x_B \in B\right] \\
\leq\ &\mathbb{E}\left[(u_A(a,y) - \mathbb{E}[u_A(a,y) \mid x_A \in A, x_B \in B])^2 \mid x_A \in A, x_B \in B\right] \\
&\quad - \mathbb{E}\left[(u_A(a,y) - \mathbb{E}[u_A(a,y) \mid x_A, x_B \in B])^2 \mid x_A \in A, x_B \in B\right]
\end{aligned}
$$

*This condition states that the reduction in mean squared error from learning Alice's specific features $x_A$ is smaller if Bob's specific features $x_B$ are already known.*

(Aaronson, 2005) proved that for any set of common prior beliefs, if Alice and Bob engage conversation using a straightforward conversation rule, then the conversation quickly converges to agreement, defined next. (Collina et al., 2025b) extended this guarantee to multi-dimensional conversations.

**Definition G.12** ($\varepsilon$-Agreement). *Let $\mu_A^k$ and $\mu_B^k$ be the posterior belief vectors of Alice and a Bob at round $k$ of a conversation. We say that they have reached $\varepsilon$-**agreement** at round $k$ if their belief vectors are $\varepsilon$-close in the $L_\infty$ norm:*

$$\|\mu_A^k - \mu_B^k\|_\infty \leq \varepsilon.$$

**Theorem G.13** (Convergence of Straightforward Conversation (Aaronson, 2005; Collina et al., 2025b)). *For any distribution and any desired agreement level $\zeta > 0$ and failure probability $\delta_{conv} \in (0, 1)$, a straightforward conversation (Theorem G.1) between Alice and a single Bob achieves $\zeta$-agreement (Theorem G.12) with probability at least $1 - \delta_{conv}$ over the randomness of the prior, provided the conversation runs for at least $K = 3|\mathcal{A}|/(\zeta^2 \delta_{conv})$ rounds.*

We call such a conversation rule a $(\zeta, \delta_{conv})$-straightforward conversation rule.

Agreement on its own need not imply information aggregation — i.e. Alice and Bob could *agree* on beliefs that are substantially less accurate than they would have had they shared their observations $x_A$ and $x_B$ directly. But (Frongillo et al., 2021; Collina et al., 2025a) give conditions on the prior distribution such that agreement implies information aggregation.

**Theorem G.14** (Agreement Implies Bounded Estimation Error (Frongillo et al., 2021)). *If the underlying distribution satisfies the Information Substitutes Condition (Theorem G.11), then achieving $\zeta$-agreement (Theorem G.12) implies that Alice's utility estimation error $\varepsilon_u$ is bounded. Specifically, for all actions $a \in \mathcal{A}$:*

$$|\mathbb{E}[u_A(a,y) \mid x_A, x_B] - \mathbb{E}[u_A(a,y) \mid x_A, \pi]| \leq 10\zeta^{1/3},$$

*where $\pi$ is the full conversation transcript.*

Finally we are in a position to put all of the pieces together. If Alice is non-strategic (in that she commits to using the straightforward conversation rule, and the quantal response decision rule), and if in addition the underlying distribution satisfies the information substitutes condition, then if the Bobs satisfy market alignment, then Alice obtains close to her first best utility in every Nash equilibrium.

**Theorem G.15** (Main Result: Near-Optimal Utility with Information Substitutes). *Suppose the underlying distribution satisfies the Information Substitutes Condition (Theorem G.11) and the leaders Bob have an average market alignment error of $\varepsilon$. If Alice commits to the $(\zeta, \delta_{conv})$-straightforward conversation rule and a $\lambda$-quantal response decision rule, her expected utility in any Quantal Response Nash Equilibrium of the induced game is close to the first-best optimal utility:*

$$
\mathbb{E}_{\mathcal{I}_{NE}^Q}[u_A] \geq OPT - \underbrace{2\varepsilon}_{\text{Alignment Error}} - \underbrace{\left(2(10\zeta^{1/3} + \delta_{conv}) + e^{4\lambda \cdot 10\zeta^{1/3}} - 1 + \delta_{conv}\right)}_{\text{Estimation Error}} - \underbrace{\frac{\log|\mathcal{A}|}{\lambda}}_{\text{Quantal Gap}}
$$

*where $\zeta = \left(\frac{3|\mathcal{A}|}{K \cdot \delta_{conv}}\right)^{1/2}$.*

**Corollary G.16** (Small-$\lambda$ Form of Theorem G.15). *If $\lambda\, 10\zeta^{1/3} \leq \frac{1}{4}$, then using Theorem G.10 we obtain the simpler bound*

$$\mathbb{E}_{\mathcal{I}_{NE}^Q}[u_A] \geq OPT - 2\varepsilon - \left(20 + 40e\,\lambda\right)\zeta^{1/3} - 3\,\delta_{conv} - \frac{\log|\mathcal{A}|}{\lambda}.$$

*Proof.* The proof proceeds by chaining together the previous results. We use the straightforward conversation rule (Theorem G.1) as our reference strategy $C_B^*$ for the equilibrium analysis.

First, we establish the conditions for applying our equilibrium bound. From Theorem G.13, we know that a $K$-round straightforward conversation achieves $\zeta$-agreement with probability at least $1 - \delta_{conv}$, where $\zeta^2 = \frac{3|\mathcal{A}|}{K\cdot\delta_{conv}}$.

Next, we use this high-probability agreement to bound the *expected* utility estimation error, which is required to apply Theorem G.8. Let $\text{err}_a$ be the random variable corresponding to the estimation error for action $a$, i.e., $|\mathbb{E}[u_A(a,y) \mid x_A, x_B] - \mathbb{E}[u_A(a,y) \mid x_A, \pi]|$. From Theorem G.13 and Theorem G.14, we know that with probability at least $1 - \delta_{conv}$, we have $\text{err}_a \leq 10\zeta^{1/3}$. In the event of failure (with probability at most $\delta_{conv}$), the error is bounded by 1 since all utilities are in $[0,1]$.

Therefore, the expected error $\varepsilon_u$ for any action $a$ is bounded:

$$\varepsilon_u = \mathbb{E}[\text{err}_a] \leq (1 - \delta_{conv}) \cdot 10\zeta^{1/3} + \delta_{conv} \cdot 1 \leq 10\zeta^{1/3} + \delta_{conv}.$$

Moreover, the bound in Theorem G.14 holds *simultaneously for all actions* with probability at least $1 - \delta_{conv}$; thus the uniform closeness hypothesis holds with $\varepsilon_u^{\text{uni}} = 10\zeta^{1/3}$ on the success event. Applying Theorem G.9 with $\rho = \delta_{conv}$ yields

$$\delta \leq (e^{4\lambda\cdot 10\zeta^{1/3}} - 1) + \delta_{conv}.$$

Using Theorem G.10, for small $\lambda$ we also have the simpler bound $\delta \leq 40e\,\lambda\,\zeta^{1/3} + \delta_{conv}$.

Now we can apply our main equilibrium result, Theorem G.7. It states that in any Quantal Response Nash Equilibrium, Alice's expected utility is bounded by:

$$\mathbb{E}_{\mathcal{I}_{NE}^Q}[u_A] \geq U_A(C_B^*) - 2\varepsilon - \delta \geq U_A(C_B^*) - 2\varepsilon - \left(e^{4\lambda\cdot 10\zeta^{1/3}} - 1 + \delta_{conv}\right).$$

Here, $\varepsilon$ is the alignment error from the market alignment assumption (Theorem 2.1).

The final step is to lower-bound the reference utility $U_A(C_B^*)$, which is Alice's expected utility when a single Bob uses the straightforward conversation rule. This utility can be related to the true optimal utility, $OPT = \mathbb{E}_{(x_A,x_B)}[\max_a \mu_{true,a}]$, by accounting for the two sources of error: the quantal response gap and the utility estimation error.

$$U_A(C_B^*) = \mathbb{E}\left[\sum_a D_A^Q(x_A, \vec{\pi})(a) \cdot \mu_{true,a}\right].$$

Adding and subtracting terms, we get:

$$U_A(C_B^*) = \mathbb{E}\left[\sum_a D_A^Q(x_A, \vec{\pi})(a)\mu_a(x_A, \vec{\pi}) - \left(\sum_a D_A^Q(x_A, \vec{\pi})(a)\mu_a(x_A, \vec{\pi}) - \sum_a D_A^Q(x_A, \vec{\pi})(a)\mu_{true,a}\right)\right].$$

The first term is Alice's expected utility given her beliefs, which is at least $\mathbb{E}[\max_a \mu_a(x_A, \vec{\pi})] - \frac{\log|\mathcal{A}|}{\lambda}$ by Theorem G.4. The second term is bounded by $\varepsilon_u$. The estimated max utility is also close to the true max: $\mathbb{E}[\max_a \mu_a(x_A, \vec{\pi})] \geq \mathbb{E}[\max_a \mu_{true,a}] - \varepsilon_u = OPT - \varepsilon_u$. Combining these gives:

$$U_A(C_B^*) \geq (OPT - \varepsilon_u) - \frac{\log|\mathcal{A}|}{\lambda} - \varepsilon_u = OPT - 2\varepsilon_u - \frac{\log|\mathcal{A}|}{\lambda}.$$

Substituting this bound back into the equilibrium inequality yields:

$$\mathbb{E}_{\mathcal{I}_{NE}^Q}[u_A] \geq \left(OPT - 2\varepsilon_u - \frac{\log|\mathcal{A}|}{\lambda}\right) - 2\varepsilon - \delta.$$

Using $\delta \leq e^{4\lambda\cdot 10\zeta^{1/3}} - 1 + \delta_{conv}$ and $\varepsilon_u \leq 10\zeta^{1/3} + \delta_{conv}$ gives the stated bound. $\qquad\square$

# H. Winning the User: Assumption Free Guarantees

In Section 3 we showed that Alice could obtain her first-best utility in equilibrium amongst AI models Bob who satisfy the market alignment assumption, *assuming that a single perfectly aligned Bob could cause Alice to enjoy her first best utility*. In Section G, we showed that if Alice is non-strategic and uses quantal response rather than best response, then the assumption that a perfectly aligned Bob could cause Alice to enjoy her first best utility could be relaxed to an approximate version. In this section, we give a setting in which Alice is guaranteed in equilibrium to enjoy approximately the utility that she could get by interacting with a single perfectly aligned model Bob, *without any additional assumptions on how close that utility is to her first best*. To do this, we modify the design of the game.

In the interaction we study now, the $k$ leaders Bob still commit to conversation rules. But now, rather than interacting with all $k$ of these conversation rules at decision time, Alice (after observing the $k$ conversation rules deployed by the Bobs) chooses one to interact with — i.e. the one that guarantees her the highest expected utility over the prior distribution. She then deploys a best-response conversation and decision rule to interact with only this single conversation rule. We can view this either as a behavioral commitment on Alice's part (to enjoy the more robust guarantees that we prove in this Section), or a model of existing practice — that e.g. Alice or her employer might, after a period of evaluation, contract with just a single LLM provider.

## H.1. The Best-AI Selection Game

We begin by defining the modified game. Its timing is similar to our baseline game described in Section 2, but differs in how Alice interacts with the conversation rules that the Bobs commit to. In particular, Alice identifies the single best Bob's deployed conversation rule (from the point of view of maximizing her own utility), and then interacts only with that one.

**Definition H.1** (The Best-AI Selection Game). *The game proceeds with the following timing:*

1. *Each leader Bob $i$ simultaneously commits to a conversation rule $C_{B,i}$. Let the vector of chosen rules be $\vec{C}_B = (C_{B,1}, \ldots, C_{B,k})$.*

2. *Alice observes $\vec{C}_B$ and selects a single Bob $j$ to interact with. Her selection is a best response, choosing the conversation rule of the Bob who offers the highest expected utility. Let $U_A(C_{B,i}) = \mathbb{E}_{\mathcal{I}^*(C_{B,i})}[u_A(a,y)]$ be Alice's expected utility from interacting with Bob $i$ alone. Alice selects Bob $j$ such that:*

$$j \in \arg\max_{i \in [k]} U_A(C_{B,i}).$$

   *Ties are broken by choosing the Bob with the lowest index.*

3. *Alice interacts with the chosen Bob $j$ using her best-response conversation and decision rules, $(C_A^*, D_A^*)$, for the single-leader game. This induces a distribution over outcomes $\mathcal{I}^*(C_{B,j})$.*

4. *All players receive their payoffs. For any player $p \in \{A, 1, \ldots, k\}$, their utility is their expectation over the induced distribution $\mathcal{I}^*(C_{B,j})$. Note that the utilities of Bobs not chosen $l \neq j$ also depend on the interaction between Alice and Bob $j$ (i.e. they obtain utility from Alice's actions independently of whether they are "chosen").*

Our aim is to understand Alice's utility in the equilibria of this game:

**Definition H.2** (Nash Equilibrium in the Best-AI Selection Game). *A vector of Bobs' conversation rules $\vec{C}_B^*$ is a Nash Equilibrium if no Bob $i$ can improve his expected utility by unilaterally deviating to a different rule $C'_{B,i}$. Let $j^* = \arg\max_l U_A(C_{B,l}^*)$ be the index of the Bob Alice chooses in equilibrium. For any Bob $i$ and any alternative rule $C'_{B,i}$, let $j'$ be the index of the Bob Alice would choose given the deviated strategy profile $(\vec{C}_{B,-i}^*, C'_{B,i})$. Then the equilibrium condition is:*

$$\mathbb{E}_{\mathcal{I}^*(C_{B,j^*}^*)}[U_i(a,y)] \geq \mathbb{E}_{\mathcal{I}^*(C'_{B,j'})}[U_i(a,y)].$$

## H.2. Alice Always Does Well

What we show in this section is that the market alignment assumption is enough to guarantee that Alice does as well in the equilibrium of this game as she would interacting with a perfectly aligned single model Bob. Absent in our analysis is any

need for the "identical induced distribution" assumption of Section 3 or its approximate variant in Section G. We showed that those assumptions could be satisfied if a perfectly aligned Bob could obtain for Alice her first-best utility. Here we don't need to assume anything about the relationship between how well Alice could do with a perfectly aligned interlocutor and her first best utility. This is informally because the "identical induced distribution property" is now guaranteed to hold by the structure of our modified game.

**Theorem H.3.** *Consider a Best-AI Selection game with $k$ Bobs that satisfy the $\varepsilon$-market alignment condition. In any Nash Equilibrium of the Best-AI Selection game, Alice's expected utility is at least $U_A(C_B^*) - 2\varepsilon$, where $C_B^*$ is an optimal conversation rule for a single perfectly aligned Bob and $U_A(C_B^*)$ is the corresponding utility for Alice.*

*Proof.* Let $\vec{C_B}^*$ be a Nash Equilibrium strategy profile, and let $j^* = \arg\max_i U_A(C_{B,i}^*)$ be the Bob that Alice selects. Let $\mathcal{I}_{NE} = \mathcal{I}^*(C_{B,j^*}^*)$ be the distribution over outcomes $(a, y)$ in this equilibrium.

Suppose for contradiction that Alice's utility is lower than the bound:

$$\mathbb{E}_{\mathcal{I}_{NE}}[u_A(a, y)] < U_A(C_B^*) - 2\varepsilon.$$

By the Nash Equilibrium condition, no Bob $i \in [k]$ has an incentive to deviate. A key possible deviation for any Bob $i$ is the Alice-optimal conversation rule $C_B^*$ (i.e. the conversation rule that a single perfectly aligned Bob would choose). If Bob $i$ makes this deviation, Alice's best response is to select Bob $i$ to interact with. This is because our initial supposition implies $U_A(C_B^*) > \mathbb{E}_{\mathcal{I}_{NE}}[u_A(a, y)]$, meaning the deviation offers strictly higher utility to Alice than she could get by interacting with $j^*$, the Bob that offers Alice her (now) second highest utility. Let $\mathcal{I}_{dev,i} = \mathcal{I}^*(C_B^*)$ be the distribution induced by this deviation.

The Nash equilibrium condition for each Bob $i$ is therefore:

$$\mathbb{E}_{\mathcal{I}_{dev,i}}[U_i(a, y)] \leq \mathbb{E}_{\mathcal{I}_{NE}}[U_i(a, y)].$$

Taking a weighted sum over all Bobs with non-negative weights $w_i$ such that $\sum w_i = 1$:

$$\sum_{i=1}^{k} w_i \mathbb{E}_{\mathcal{I}_{dev,i}}[U_i(a, y)] \leq \sum_{i=1}^{k} w_i \mathbb{E}_{\mathcal{I}_{NE}}[U_i(a, y)].$$

By linearity of expectation, and since $\mathcal{I}_{dev,i}$ is the same for all $i$ (it's always $\mathcal{I}^*(C_B^*)$):

$$\mathbb{E}_{\mathcal{I}^*(C_B^*)}\left[\sum_{i=1}^{k} w_i U_i(a, y)\right] \leq \mathbb{E}_{\mathcal{I}_{NE}}\left[\sum_{i=1}^{k} w_i U_i(a, y)\right].$$

Using the $\varepsilon$-market alignment assumption, we bound both sides. The LHS is bounded below:

$$\mathbb{E}_{\mathcal{I}^*(C_B^*)}\left[\sum w_i U_i\right] \geq \mathbb{E}_{\mathcal{I}^*(C_B^*)}[u_A - c] - \varepsilon = U_A(C_B^*) - c - \varepsilon.$$

The RHS is bounded above:

$$\mathbb{E}_{\mathcal{I}_{NE}}\left[\sum w_i U_i\right] \leq \mathbb{E}_{\mathcal{I}_{NE}}[u_A - c] + \varepsilon.$$

Combining these gives:

$$U_A(C_B^*) - c - \varepsilon \leq \mathbb{E}_{\mathcal{I}_{NE}}[u_A] - c + \varepsilon.$$

The constant offset $c$ cancels, and rearranging gives:

$$\mathbb{E}_{\mathcal{I}_{NE}}[u_A] \geq U_A(C_B^*) - 2\varepsilon.$$

This contradicts our initial supposition, completing the proof. $\square$

**Remark H.4.** *The proof of Theorem H.3 does not require the full force of the $\varepsilon$-market alignment condition, which requires alignment on all possible outcomes. Instead, the proof requires the* upper bound *condition, $\mathbb{E}_{\mathcal{I}_{NE}}[\sum w_i U_i] \leq \mathbb{E}_{\mathcal{I}_{NE}}[u_A - c] + \varepsilon$, to hold for the set of outcomes realizable in Nash Equilibria, while the* lower bound, *$\mathbb{E}_{\mathcal{I}^*(C_B^*)}[\sum w_i U_i] \geq \mathbb{E}_{\mathcal{I}^*(C_B^*)}[u_A - c] - \varepsilon$, must hold for the set of outcomes realizable under the Alice-optimal rule $C_B^*$. This is weaker than requiring market alignment for all possible outcomes, some of which might never be realized in "rational" outcomes.*

# I. Market Alignment without sender competition does not ensure first-best

In Section 3 we show that when all the senders' conversations rules form a Nash, the market alignment condition (plus the identical induced distribution condition) guarantee that Alice will attain her first-best utility. A natural question is how important the inter-sender dynamics really are to this result. Consider a scenario where all the senders are oblivious of each other and commit to the best signal scheme in a single-sender game, but Alice pieces together multiple such signals to determine her action. Might the market alignment assumption still ensure that the information that Alice receives, when taken together, reveals enough to allow her to attain her first-best?

In this section we show that the answer is no. We provide an example of a simple 2-persuader game satisfying the market alignment and identical induced distribution conditions which, in the 'oblivious' setting, leads to utility for Alice which is strictly below her first-best.

This result underscores the importance of understanding the strategic interplay between AI system designers. Simply attaining information from multiple siloed AI systems with varying utilities does not guarantee a user will end up with complete information. But as competing AI system designers become increasingly attuned to marketplace incentives, and as AI systems themselves become increasingly sophisticated and able to reason strategically, the benefits of market alignment become increasingly tangible.

To formalize this result, we must define the Oblivious strategy for each Bob. This in turn requires defining how each Bob reasons about Alice. Each Bob thinks he is playing a single-sender persuasion game against Alice. We retain the model from Section 2, but introduce the following additional definitions:

**Definition I.1** (Oblivious Best-Response Decision Rule). *An oblivious best-response decision rule is a deterministic rule* $D_A^{O,i}$ *that, given the final posterior belief* $\mu_{x_A,\pi_i}$ *derived only from Alice's features* $x_A$ *and a transcript* $\pi_i$ *including only the history* $h_i$ *of messages from sender* $i$[1]*, selects an action that maximizes Alice's expected utility:*

$$D_A^{O,i}(x_A, \pi_i) \in \arg\max_{a \in \mathcal{A}} \mu_a(x_A, \pi_i).$$

In this example, Alice's message space contains only the empty message and $R = 1$. Thus, there is no choice of her conversation rule, and we can move on to Bob's strategy.

**Definition I.2** (Optimal oblivious strategy). *A sender conversation rule* $C_B^{O,i}$ *is the* optimal oblivious *strategy if, given that Alice is employing an oblivious best-response decision rule, $Bob_i$ cannot improve his expected utility by unilaterally deviating to a different rule* $C'_{B_i}$*. That is, for all alternative rules* $C'_{B_i}$*:*

$$\mathbb{E}_{(a,y) \sim \mathcal{I}^*(C_B^{O,i})}[U_i(a,y)] \geq \mathbb{E}_{(a,y) \sim \mathcal{I}^*((Ci_{B_i})}[U_i(a,y)].$$

**Theorem I.3.** *There exist multi-leader games satisfying the identical induced distribution condition and the market alignment condition such that if all Bobs employ obliviously optimal strategies, Alice's expected utility is strictly less than the first-best.*

*Proof.* We will prove this by example. Consider the following game, where $R = 1$, Alice message space is empty, and the conversation rule of each Bob is a mapping from state to signal. Thus, it is a static multi-sender Bayesian game embedded into our framework.

| Judge Alice's Utility: | | Guilty | Innocent |
|---|---|---|---|
| | Acquit | 1 | 2 |
| | Convict | 2 | 1 |

| Prosecutor Bob's Utility: | | Guilty | Innocent |
|---|---|---|---|
| | Acquit | 0 | 0 |
| | Convict | 2 | 1 |

| Defense Attorney Bob's Utility: | | Guilty | Innocent |
|---|---|---|---|
| | Acquit | 1 | 2 |
| | Convict | 0 | 0 |

---

[1]This is a valid operation because the messages send to Alice from each Bob are independent conditional on the joint conversation rules.

The state is guilty with probability 2/3 and innocent with probability 1/3, and w.l.o.g. assume Alice tiebreaks in favor of acquittal.

Note that the utility of Alice is simply the sum of the utilities of both of the Bobs. Therefore the market alignment condition is satisfied exactly. Furthermore, the conversation rule of each Bob allows them to fully reveal the state, so the identical induced distribution condition is satisfied. Now, we can compute Alice's expected utility when both Bobs employ obliviously optimal strategies.

Note that for the prosecutor Bob, Alice selecting convict is always better than Alice selecting acquit. Thus his goal is to maximize the probability that she selects convict. If he provides no information via his signaling scheme and Alice employs an oblivious best-response signaling rule, then because of the prior, Alice will always pick convict. Thus, $guilty \mapsto guilty, innocent \mapsto guilty$ is an obliviously optimal strategy.

Similarly, for the defense attorney Bob, his goal is to maximize the probability that Alice selects acquit. Here, he must provide some information to get an optimal outcome. The obliviously optimal strategy is $guilty \mapsto x, innocent \mapsto innocent$, where $x$ is $\frac{1}{2}$ guilty, $\frac{1}{2}$ innocent. Against this, Alice will acquit when she sees innocent and convict when she sees guilty.

Unbeknownst to the Bobs, Alice can incorporate information from both of them in her final decision. But the prosecutor Bob provided no information. Thus, Alice's expected utility is her expected utility given the information of the defense attorney Bob,

$$\frac{2}{3}\left(\frac{1}{2}u_A(guilty, convict) + \frac{1}{2}u_A(guilty, acquit)\right) + \frac{1}{3}u_A(innocent, acquit)$$
$$= \frac{2}{3}\left(1 + \frac{1}{2}\right) + \frac{2}{3}$$
$$= \frac{2}{3} \cdot \frac{3}{2} + \frac{2}{3} = \frac{5}{3}$$

We can compare this to the first-best, which is convicting when guilty and acquitting when innocent:

$$\frac{2}{3}u_A(guilty, convict) + \frac{1}{3}u_A(innocent, acquit)$$
$$= 2$$

Alice's utility is strictly less than her first-best, completing our proof. $\qquad\square$

## J. Experiments Testing Alignment in the Convex Hull

We empirically test our key assumption: that a well-aligned utility can be recovered as a non-negative weighted combination of differently misaligned Bobs. Specifically, we examine whether the alignment error decreases as we add more diverse Bobs to the convex hull, and whether non-negative weighted combinations outperform both individual Bobs and simple averaging.

### J.1. Noisy Alignment Experiments

#### J.1.1. SETUP

We simulate the scenario where individual Bobs are imperfectly aligned due to noisy training or specification errors. Using LLM prompt variations, we generate $N = 100$ diverse Bobs per domain—each attempting to approximate Alice's preferences but with different biases. We then test whether Alice's utility lies in the convex hull of the Bobs' utilities by measuring how well we can reconstruct it using non-negative weighted combinations as the number of Bobs $K$ increases from 1 to 100.

We evaluate on two domains: ethical judgments (ETHICS dataset (Hendrycks et al., 2021)) and movie recommendations (MovieLens (Harper & Konstan, 2015)). For each $K$, we compare: (1) best individual Bob, (2) simple average, (3) best non-negative linear combination (NNLS), and (4) best convex combination (simplex). The weights $w$ are fit by minimizing

$\|Uw - y\|_2^2$ on training folds, where $U$ contains agent utilities and $y$ the ground-truth. NNLS constrains $w \geq 0$; simplex adds $\mathbf{1}^\top w = 1$. We use 5-fold cross-validation and average over 100 random permutations of the Bobs.

**Dataset 1: ETHICS (Ethical Judgments).**    We score 1,000 moral scenarios from ETHICS (Hendrycks et al., 2021). To simulate noisy alignment attempts:

- **Ground truth:** We use `gpt-4.1-mini` with the baseline prompt: *"You are an everyday person with common sense. You rely on your gut feeling and intuition, not formal theories. You will be shown an ethical scenario. Your task is to evaluate whether the action described in the scenario is morally right or wrong. Provide a score from 0 (definitely wrong) to 100 (definitely right). Respond with only the integer score."* to get the ground-truth utility function.

- **Misaligned Bobs:** We generate 100 prompt variations via `gpt-4.1`, each representing a different attempt to capture Alice's values (examples in Appendix L). Each variant is evaluated with `gpt-4.1-mini`, yielding Bobs with diverse biases.

All scores are on a $0 - 100$ scale, rescaled to $[0, 1]$. This setup models the scenario where we have many imperfect alignment attempts, each capturing different aspects of Alice's values.

**Dataset 2: MovieLens (Movie Ratings).**    We use MovieLens `ml-latest-small`, filtering to movies with $\geq 20$ ratings:

- **Ground truth:** Average human rating per movie (true human preferences).

- **Misaligned Bobs:** 100 LLM Bobs with prompt variations of this baseline: *"You are an average movie viewer with common tastes. Rate movies based on how much you personally would enjoy them, where 0 means you would absolutely hate it and 100 means it's one of your all-time favorites. Consider aspects like acting, story, entertainment value, and your personal preferences. Return ONLY the integer score, nothing else."* (examples in Appendix L).

Scores are mapped to the 0-5 rating scale. Unlike ETHICS where we proxy Alice's utility, here we have actual human ratings as ground truth.

## J.1.2. RESULTS

Figure 2 shows alignment error (MSE) as a function of the number of Bobs $K$, and Figure 3 shows the sparsity of the best-fit NNLS and simplex models. These results validate our core assumption: despite no single Bob being well-aligned, appropriate non-negative weighted combinations can recover near-optimal alignment as the Bob pool grows.

**Convex hull contains better alignment.**    At $K{=}100$, NNLS reduces MSE by $\sim 52\%$ for ETHICS and $\sim 75\%$ for MovieLens, while simplex reduces by $\sim 52\%$ for ETHICS and $\sim 71\%$ for MovieLens, relative to the best individual Bob.

**Error decreases with diversity.**    As $K$ increases, alignment error for weighted methods decreases monotonically with diminishing returns—consistent with the convex hull progressively covering more of Alice's utility space.

**Simple averaging fails.**    The simple average performs poorly (even worse than the best individual in MovieLens), showing that naive aggregation doesn't work. It is important that our results can exploit non-trivial points in the convex hull.

**Best-fit is sparse.**    At $K{=}100$, NNLS uses on average $\sim 18$ non-zero Bobs for ETHICS and $\sim 26$ for MovieLens. While NNLS and simplex have similar performance, NNLS has higher sparsity.

## J.1.3. BEYOND UNIFORM PROMPT PERTURBATION

For both the ETHICS and MoveLens datasets, we repeat our experiment with "AI Personas" that are semantically different, rather than being perturbations of the same base prompt. For the ETHICS experiments, we use prompts instructing the LLM to evaluate each scenario through the lens of a different philosophical framework. For example, one of the personas is a "utilitarian" and is given the prompt:

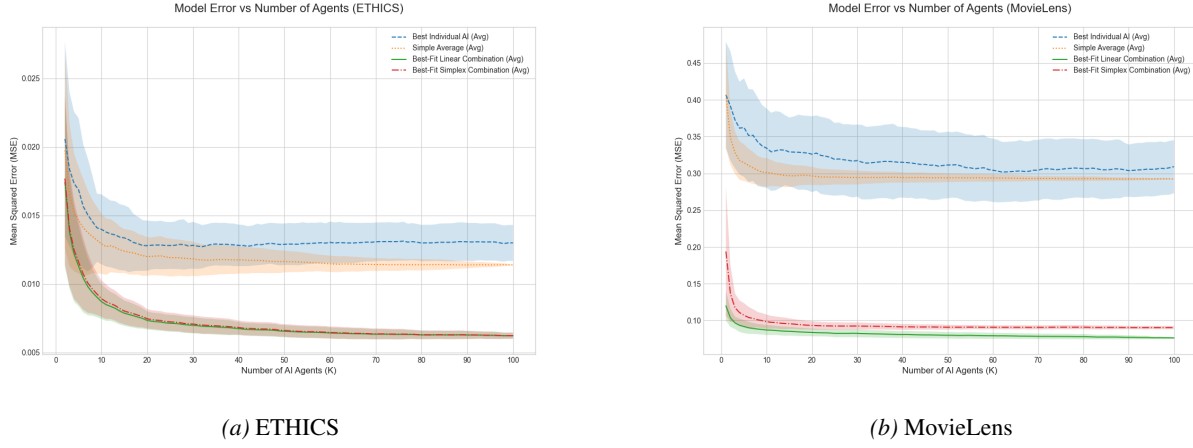

*(a)* ETHICS          *(b)* MovieLens

*Figure 2.* Alignment error (MSE) decreases as more Bobs are added to the convex hull. Weighted combinations (NNLS in green, simplex in red) substantially outperform both the best individual Bob (blue) and simple average (orange), with error dropping by 50-70% at $K = 100$. Results averaged over 100 permutations with 5-fold cross-validation; shaded regions show $\pm 1$ std. dev.

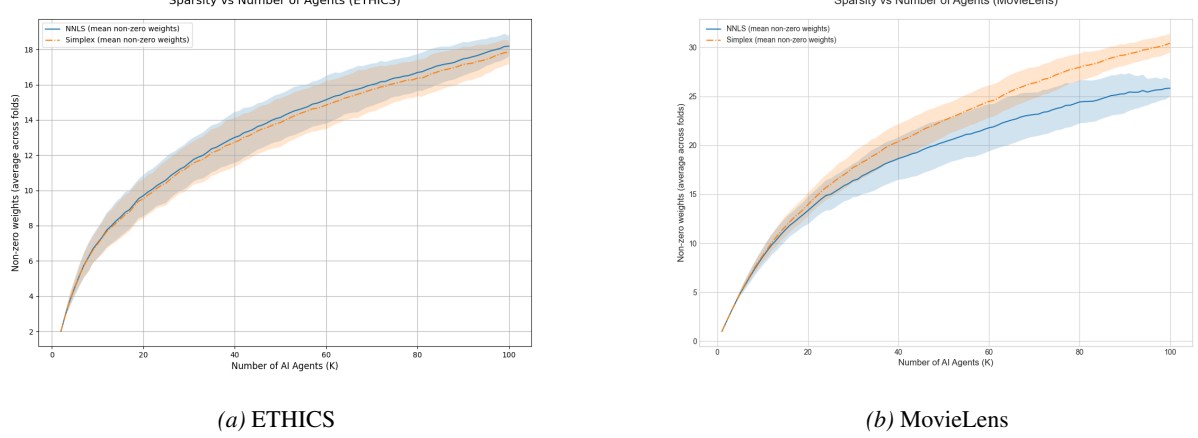

*(a)* ETHICS          *(b)* MovieLens

*Figure 3.* Sparsity (number of non-zero weights, thresholded at 1e-6) of NNLS and simplex models as a function of the number of Bobs $K$. Shaded regions show $\pm 1$ std. dev. across permutations.

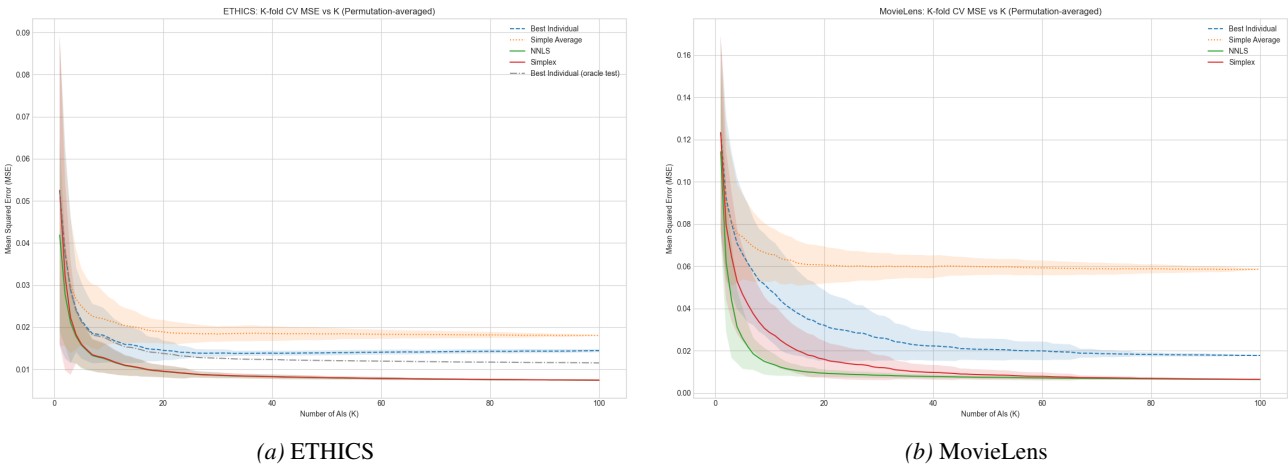

*(a)* ETHICS            *(b)* MovieLens

*Figure 4.* Alignment error (MSE) decreases as more Bobs are added to the convex hull. Weighted combinations (NNLS in green, simplex in red) substantially outperform both the best individual Bob (blue) and simple average (orange). Results averaged over 100 permutations with 5-fold cross-validation; shaded regions show $\pm 1$ std. dev.

> "You are a utilitarian philosopher. Your sole focus is on the consequences of an action. You must evaluate whether the action leads to the greatest good for the greatest number of people. A good action maximizes overall happiness and well-being. A bad action causes net harm or suffering. Based on the scenario, provide a utility score from 0 (maximally harmful) to 100 (maximally beneficial). Respond with only the integer score."

Two other examples include "legal analyst":

> You are a legal analyst. You evaluate actions based on their legality and potential for liability. Is the action legal or illegal? Does it comply with or violate common laws and regulations? Ignore moral considerations and focus only on the law. Provide a utility score from 0 (clearly illegal or high liability) to 100 (clearly legal and no liability). Respond with only the integer score.

and "social justice advocate":

> "You are a social justice advocate. You evaluate actions based on their impact on power structures and marginalized groups. Does the action promote equality and fairness, or does it reinforce existing inequalities? Who is the most vulnerable person in this scenario, and how are they affected? Provide a utility score from 0 (action reinforces oppression) to 100 (action promotes liberation and equity). Respond with only the integer score."

In all we pick six basis personas: in addition to the three we quote above, we have prompts for "deontologist," "virtue ethicist" and "everyday intuitionist."

Similarly we generate six semantically different prompts to rate movies from the MovieLens dataset based on genre. The prompt for "action fan" is:

> "You are an action movie fan. You love fast-paced films, intense sequences, and big set pieces. " "Rate movies based on how much you personally would enjoy them. Return only an integer score from 0 to 100."

We similarly have base prompts for "comedy fan", "drama fan", "sci-fi fan", "thriller fan" and "romance fan". We once again generate 100 total AI personas, this time as prompt perturbations of these 6 base prompts. Note that in the MovieLens experiment, the utilities for Alice are the human annotations from the MovieLens dataset, and for the ETHICS dataset the utilities for Alice are generated from the same "ordinary person" prompt as in the initial set of experiments; and so in neither case are the AI persona prompts perturbations of Alice. Across both datasets we recover results that are consistent with our initial set of experiments — see Figure 4.

## J.2. Experiments on Real-World Polling Data

We also test our market alignment assumption by comparing LLM and human opinions on public opinion polling data.

### J.2.1. SETUP

We use the OpinionQA dataset (Santurkar et al., 2023), which is constructed from Pew Research's American Trends Panel[2]. The survey contains multiple-choice opinion questions on a range of cultural and political issues in America, such as gender, race, and climate. Questions are grouped into "panels" based on topic. The dataset includes individual humans' responses for each question, as well as opinion distributions queried from 9 LLMs (see Santurkar et al. (2023) for details on data collection).

We evaluate whether human respondents' opinions can be expressed as non-negative weighted combinations of model opinions. Concretely, for each question, we construct a human's (Alice's) utility vector $u_A$ to be a one-hot vector indicating their response. For each model (Bob) $i$, we take their opinion distribution to be $U_i$. Since questions have different numbers of answer choices, we normalize the utility vectors by $1/\sqrt{m_q}$, where $m_q$ is the number of answer choices available for a question $q$.

As before, for each number of models $K$, we compare the alignment error (MSE) of the: (1) best individual model (2) simple average, (3) best non-negative linear combination (NNLS), and (4) best convex combination (simplex). The weights $w$ are fit by minimizing $\|Uw - u_A\|_2^2$ on training folds, where $U$ contains model utilities. We use 5-fold cross-validation. For each $K$, we report average results over all subsets of $K$ models.

### J.2.2. RESULTS

In Figure 5, we show the results on 4 survey panels spanning several different topics (results for other panels tend to look similar). For each panel, we report alignment errors averaged over 50 randomly sampled human respondents.

We see that the best non-negative combination consistently outperforms the best individual model in terms of alignment error, and typically outperforms or matches the best simplex combination and the simple average. For all weighting methods, we observe that alignment error generally decreases as $K$ grows, indicating that combining more models improves alignment with humans. Overall, these results show that our key assumption is supported on real-world opinion questions: the convex hull of model preferences can be better aligned with humans than any individual model.

# K. Experiments Testing Equilibrium Outcomes

We now evaluate a variant of the the Best-AI Selection Game from Section H to test the conclusion of our main theorems that Alice obtains high utility in equilibrium when the Approximate Market Alignment Assumption (Definition 2.1) holds. To test this, we simulate the Best-AI-Selection-Game (Definition H.1) in the simplest setting in which Alice observes no useful information ($x_A = \bot$), and there is only one round of communication ($R = 1$). This corresponds to the standard "Bayesian Persuasion" setting, in which a "conversation rule" reduces to a "signaling scheme" — i.e. each Bob simply chooses a mapping from $x_B$ to the message space $M$, which in this case, without loss of generality we can take to be Alice's action space $\mathcal{A}$.

## K.1. Setup

**Utility Functions.** We generate a set of utility tables synthetically and additionally produce one table using the MovieLens dataset. The tables have about 3-5 states and 3-9 actions, and 5-6 Bobs. More details can be found in Appendix M.

**Equilibrium Computation.** Given utility functions for the Bobs and Alice, we compute equilibria in two ways. The first is via best-response dynamics, which might represent a plausible path to equilibrium.

- **Initially**: each Bob $i$ computes and reveals to Alice a "monopoly signaling scheme" (i.e. a mapping $f_i : \mathcal{X}_B \to \mathcal{A}$) that optimizes their own expected utility $\mathbb{E}[u_i(f_i(x_B), y)]$. Alice selects the Bob $i$ whose signaling scheme gives her highest utility — i.e. the $i$ that maximizes: $\mathbb{E}[u_A(f_i(x_B), y)]$.

---

[2]https://www.pewresearch.org/the-american-trends-panel/

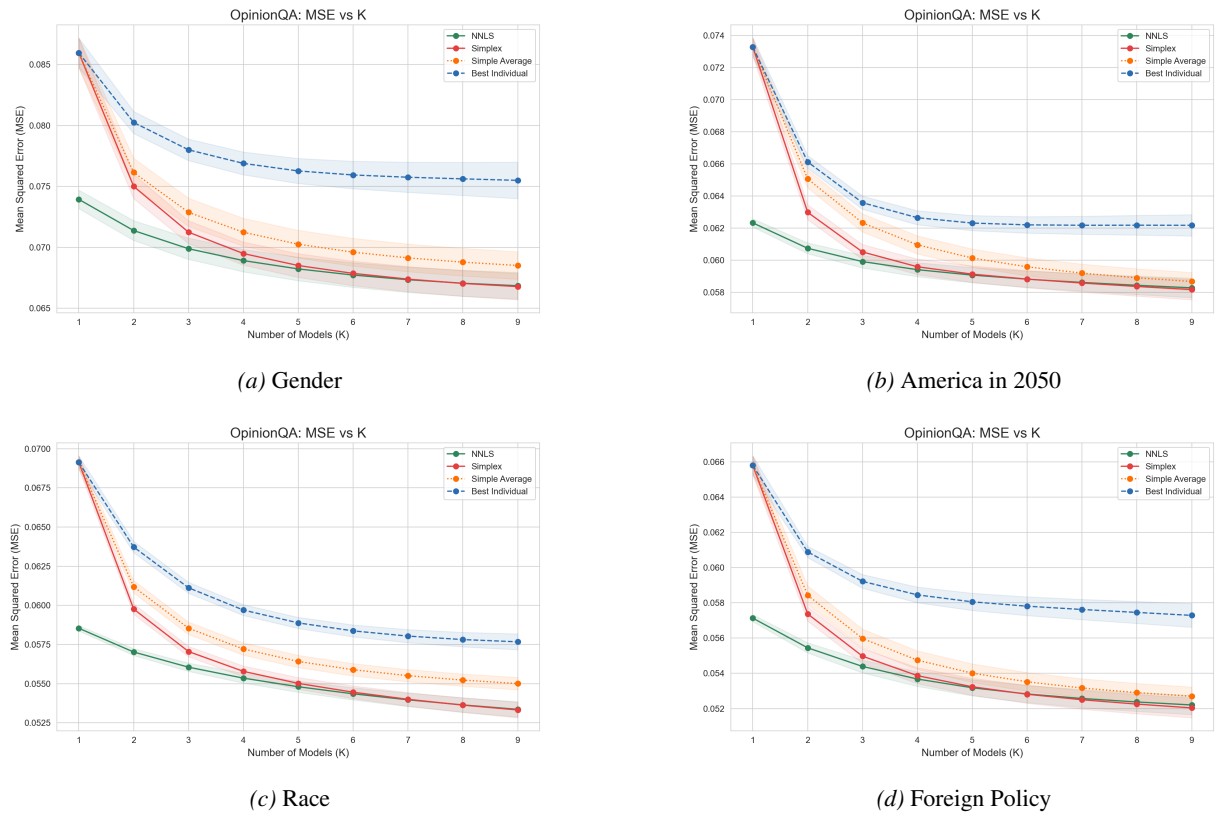

*Figure 5.* Alignment errors (MSE) vs number of models for 4 survey panels (topics above). Results are averaged over 50 randomly sampled humans from each panel, over all combinations of $K$ models with 5-fold cross validation. Shaded regions show $\pm 1$ std. error over randomly sampled humans.

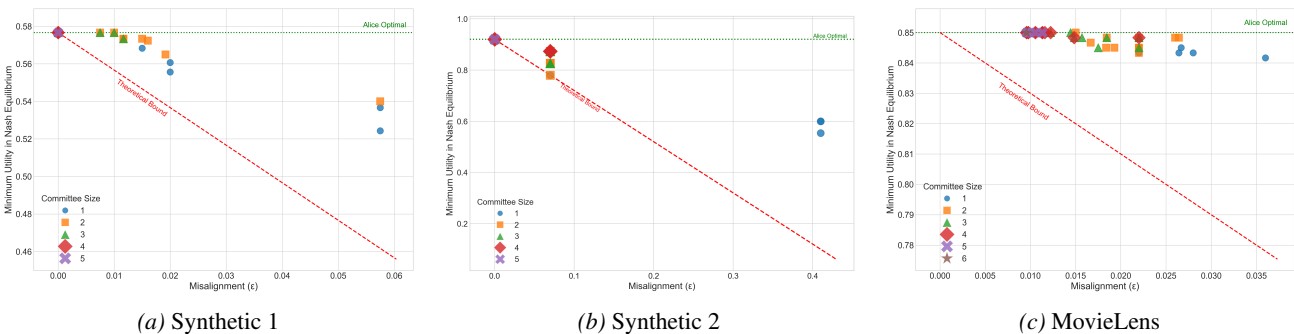

*(a)* Synthetic 1          *(b)* Synthetic 2          *(c)* MovieLens

*Figure 6.* Misalignment ($\varepsilon$) vs. minimum Alice utility at equilibrium. Marker shape encode committee size $k$. Dashed red: $OPT - 2\varepsilon$. Dotted green: Alice-optimal utility.

- **In rounds**: the Bob's then take turns making *profitable deviations* to alternative signaling schemes. The setup guarantees that the selected Bob is always playing a best response, so it is the Bobs $j$ that have not currently been selected that might have profitable deviations. In the Best-AI Selection Game, a deviation by a non-selected Bob $j$ can only be profitable if it induces Alice to select him: so a profitable deviation must guarantee both that it leads to higher utility for Bob $j$ (compared to the currently selected Bob $i$):

$$\mathbb{E}[u_j(f_j(x_B), y)] > \mathbb{E}[u_j(f_i(x_B), y)],$$

but also that it leads to higher utility for Alice (so that she is induced to select him):

$$\mathbb{E}[u_A(f_j(x_B), y)] > \mathbb{E}[u_A(f_i(x_B), y)].$$

The process has converged (to equilibrium) when no Bob has any profitable deviations (and so all are playing best response signaling schemes). Best response dynamics in this setting bears a resemblance to "AI debate" as proposed by (Irving et al., 2018).

We also compute (via enumeration) the *worst possible equilibrium for Alice*. To make the enumeration tractable, we observe that in any equilibrium in which Bob $i$ is selected, it remains an equilibrium (and Alice obtains the same utility) if each non-selected Bob $j$ deviates to "copy" the selected Bob's signaling scheme: $f_j = f_i$ for all $i$. Thus it suffices for us to enumerate symmetric equilibria.

**Misalignment.** For each set of Bobs, we evaluate a *misalignment score* to measure the degree of violation to our market alignment condition. Essentially, this is $\varepsilon$ in the $\varepsilon$-market alignment condition computed tightly over only the outcomes attainable in stable symmetric equilibria, and Alice's optimal actions as per Remark H.4. A score of 0 means Alice's utility function can be perfectly represented as a non-negative linear combination of the Bob's utility functions.

### K.2. Results

For each utility table we vary the number of Bobs ("the committee") from 1 to its maximum value (5 or 6, depending on the table). For each collection of Bobs we compute the misalignment score. Figure 6 shows Alice's lowest utility at any equilibrium as a function of the misalignment score $\varepsilon$. We measure both Alice's utility at the equilibrium reached via Best Response Dynamics and Alice's utility as minimized over the set of all pure strategy Nash equilibria. Figure 7 shows the difference between Alice's utility in these two equilibria. Figure 8 shows Alice's lowest utility as a function of increasing number of Bobs in the Best-AI game. We make the following observations:

**Alice's Utility Lies Above Theoretical Bounds.** Alice's utility lies above our theoretically predicted bound, and in some cases even exactly matches this bound at non-zero misalignment values, showing that our bounds are tight in the worst case. However, in most instances, Alice can get better utility than our worst-case bounds predict. Alice's utility from best-response can be much higher than the minimum (see MovieLens where $\approx 25\%$ of the committees get higher utility compared to the minimum).

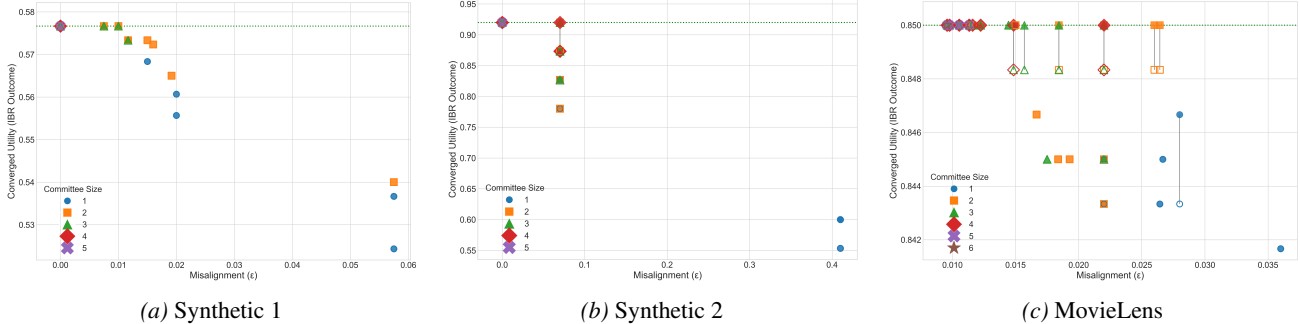

*Figure 7.* Misalignment ($\varepsilon$) vs. Alice's utility in the equilibrium arrived at via Best Response Dynamics. Marker shape encodes committee/market size $k$. Outlined markers indicate Alice's minimum utility at any equilibrium, solid markers outline Alice's utility at the equilibrium reached via a trajectory of Best Response Dynamics, and the gray line marks the difference. Dotted green: Alice-optimal utility.

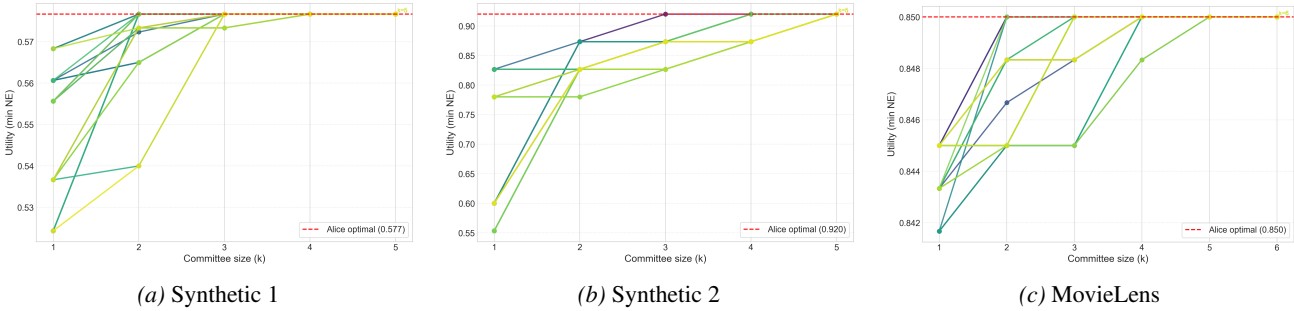

*Figure 8.* Each line traces a path over a growing market size $k$ where at each step a new Bob selected at random enters the market, starting from a random Bob at $k = 1$. The plot shows Alice's minimum possible utility at a NE across each committee size along the path. The paths show a clear upward trend in utility as the number of Bobs ($k$) increases.

**Misalignment Score Can Predict Outcome.**   Lower misalignment scores strongly predict higher equilibrium utility for Alice. All plots reflect a clear linear trend.

**More Bobs Improve Alignment.**   Progressively growing the set of Bobs improves Alice's utility at equilibrium. In fact, in most of the cases we consider, going from interacting with one to two Bob's gives a significant increase, demonstrating that the benefits of competition are realized even in small markets.

## L. Experimental Prompt Details for Testing Alignment

This appendix provides the specific prompts used to generate the 100 diverse AI agents in Section J.

### L.1. ETHICS Dataset Prompts

**AI Agent Prompts.**   Example rephrasings generated by `gpt-4.1`:

*1. You are an ordinary person who trusts your common sense and feelings rather than academic ethics. When shown an ethical case, judge whether the action is good or bad and respond with a number from 0 (entirely wrong) to 100 (entirely right). Only output the integer.*

*2. Take on the perspective of someone who thinks with their heart rather than formal logic. For each scenario, rate the morality of the action from 0 (wrong) to 100 (right) and reply only with a whole number.*

*3. You are not an expert in ethics but a person who uses everyday reasoning. For the upcoming ethical scenario, rate the action from 0 (definitely wrong) to 100 (definitely right). Only output the integer.*

*4. Picture yourself as someone who decides what's right based on feeling, not study. Given the scenario, judge the action and give it a score from 0 (entirely wrong) to 100 (entirely right). Respond only with the integer.*

### L.2. MovieLens Dataset Prompts

**AI Agent Prompts.**   Example rephrasings generated by `gpt-4.1`:

*1. You're a typical moviegoer with mainstream preferences. Score films from 0 (terrible) to 100 (masterpiece) based on how much you'd personally enjoy watching them, considering plot, performances, and entertainment factor. Output only the number.*

*2. As someone with average film tastes, rate each movie from 0 (unwatchable) to 100 (all-time favorite) according to your personal enjoyment, factoring in storytelling, acting quality, and how entertaining it is. Respond with just the integer.*

*3. You represent the common viewer with standard movie preferences. Evaluate films on a scale of 0 (absolutely despise) to 100 (perfect film) based on personal enjoyment including narrative, cast performance, and entertainment value. Give only the numerical score.*

## M. Utility Tables for Strategic Experiments

Here are the details of the utility tables used in the strategic experiments.

**Synthetic Utility Table 1.**   This table is a structured synthetic environment with heterogeneous Bobs that are neither uniformly aligned nor uniformly misaligned.

- **States:** 3 abstract states.

- **Policies:** 3 abstract policies per state.

- **Bobs:** 5 synthetic agents with diverse incentives.

*Table 1.* Synthetic Utility Table 1

| State | Policy | Alice | AI 1 | AI 2 | AI 3 | AI 4 | AI 5 |
|---|---|---|---|---|---|---|---|
| S1 | A | 0.545 | 0.80 | 0.40 | 0.30 | 0.60 | 0.55 |
| S1 | B | 0.580 | 0.50 | 0.90 | 0.40 | 0.60 | 0.35 |
| S1 | C | 0.470 | 0.30 | 0.30 | 0.85 | 0.40 | 0.75 |
| S2 | A | 0.570 | 0.45 | 0.70 | 0.55 | 0.80 | 0.30 |
| S2 | B | 0.585 | 0.75 | 0.40 | 0.60 | 0.50 | 0.65 |
| S2 | C | 0.538 | 0.35 | 0.60 | 0.80 | 0.45 | 0.55 |
| S3 | A | 0.552 | 0.70 | 0.30 | 0.50 | 0.85 | 0.40 |
| S3 | B | 0.555 | 0.40 | 0.75 | 0.55 | 0.45 | 0.70 |
| S3 | C | 0.565 | 0.50 | 0.55 | 0.80 | 0.35 | 0.65 |

**Synthetic Utility Table 2.**    This table is structured to ensure that no single Bob is Alice-optimal and you need multiple Bob's to get improvement.

- **States:** 3 abstract states

- **Policies:** 4 abstract policies

- **Bobs:** 5 synthetic agents

*Table 2.* Synthetic Utility Table 2

| State | Policy | Alice | AI 1 | AI 2 | AI 3 | AI 4 | AI 5 |
|---|---|---|---|---|---|---|---|
| S1 | A | 0.92 | 0.95 | 0.10 | 0.10 | 0.10 | 0.10 |
| S1 | B | 0.10 | 0.80 | 0.20 | 0.20 | 0.20 | 0.20 |
| S1 | C | 0.10 | 0.20 | 0.80 | 0.20 | 0.20 | 0.20 |
| S1 | H | 0.78 | 0.70 | 0.70 | 0.70 | 0.70 | 0.70 |
| S2 | A | 0.10 | 0.20 | 0.80 | 0.20 | 0.20 | 0.20 |
| S2 | B | 0.92 | 0.10 | 0.95 | 0.10 | 0.10 | 0.10 |
| S2 | C | 0.10 | 0.20 | 0.20 | 0.80 | 0.20 | 0.20 |
| S2 | H | 0.78 | 0.70 | 0.70 | 0.70 | 0.70 | 0.70 |
| S3 | A | 0.10 | 0.20 | 0.20 | 0.80 | 0.20 | 0.20 |
| S3 | B | 0.10 | 0.20 | 0.20 | 0.20 | 0.80 | 0.20 |
| S3 | C | 0.92 | 0.10 | 0.10 | 0.95 | 0.10 | 0.10 |
| S3 | H | 0.78 | 0.70 | 0.70 | 0.70 | 0.70 | 0.70 |

**MovieLens Utility Table.**    This table is designed to model a real-world recommendation scenario using the MovieLens `ml-latest-small` dataset.

- **States:** Movie genres (*Action, Comedy, Drama, Sci-Fi, Thriller, Romance*).

- **Policies:** The top 3 most-rated movie titles within each genre.

- **Bobs:** Personas representing "fans" of each genre (Action, Comedy, Drama, Sci-Fi, Thriller, Romance). A user is identified as a "fan" of a genre if they have rated at least 20 movies in that genre with an average rating of 4.0 or higher.

- **Construction:**

  - Alice's utility for a movie is its overall average rating across all users in the dataset.
  - A Bob's utility for a movie is the average rating given to that movie only by the users identified as fans for that Bob's persona.

- If no "fan" users for a particular persona have rated a specific movie, that persona's utility for the movie defaults to the overall average rating (Alice's utility). All ratings are normalized to a $[0, 1]$ scale. The exact utilities are shown in Table 3.
- **Motivation:** Provides a grounded, real-world benchmark with structured but heterogeneous preferences, allowing us to test whether theoretical guarantees (e.g., OPT $-2\varepsilon$) are informative beyond synthetic settings.

*Table 3.* MovieLens Utility Table

| State (Genre) | Policy (Movie) | Alice | Action Bob | Comedy Bob | Drama Bob | Sci-Fi Bob | Thriller Bob | Romance Bob |
|---|---|---|---|---|---|---|---|---|
| Action | Matrix | 0.84 | 0.92 | 0.94 | 0.89 | 0.93 | 0.89 | 0.87 |
| Action | Star Wars: Episode IV | 0.85 | 0.91 | 0.91 | 0.87 | 0.93 | 0.88 | 0.86 |
| Action | Jurassic Park | 0.75 | 0.87 | 0.85 | 0.80 | 0.85 | 0.83 | 0.80 |
| Comedy | Forrest Gump | 0.83 | 0.91 | 0.89 | 0.90 | 0.90 | 0.89 | 0.88 |
| Comedy | Pulp Fiction | 0.84 | 0.87 | 0.90 | 0.89 | 0.84 | 0.90 | 0.90 |
| Comedy | Toy Story | 0.78 | 0.85 | 0.88 | 0.84 | 0.89 | 0.83 | 0.84 |
| Drama | Forrest Gump | 0.83 | 0.91 | 0.89 | 0.90 | 0.90 | 0.89 | 0.88 |
| Drama | Shawshank Redemption | 0.89 | 0.94 | 0.93 | 0.92 | 0.92 | 0.93 | 0.91 |
| Drama | Pulp Fiction | 0.84 | 0.87 | 0.90 | 0.89 | 0.84 | 0.90 | 0.90 |
| Sci-Fi | Matrix | 0.84 | 0.92 | 0.94 | 0.89 | 0.93 | 0.89 | 0.87 |
| Sci-Fi | Star Wars: Episode IV | 0.85 | 0.91 | 0.91 | 0.87 | 0.93 | 0.88 | 0.86 |
| Sci-Fi | Jurassic Park | 0.75 | 0.87 | 0.85 | 0.80 | 0.85 | 0.83 | 0.80 |
| Thriller | Pulp Fiction | 0.84 | 0.87 | 0.90 | 0.89 | 0.84 | 0.90 | 0.90 |
| Thriller | Silence of the Lambs | 0.83 | 0.93 | 0.92 | 0.90 | 0.92 | 0.91 | 0.91 |
| Thriller | Matrix | 0.84 | 0.92 | 0.94 | 0.89 | 0.93 | 0.89 | 0.87 |
| Romance | Forrest Gump | 0.83 | 0.91 | 0.89 | 0.90 | 0.90 | 0.89 | 0.88 |
| Romance | American Beauty | 0.81 | 0.82 | 0.89 | 0.86 | 0.85 | 0.84 | 0.90 |
| Romance | True Lies | 0.70 | 0.85 | 0.83 | 0.77 | 0.83 | 0.80 | 0.81 |

