# OpenReview forum: "Emergent Alignment via Competition"
_ICML.cc/2026/Conference — ICML 2026 regular_

### Official Review · Reviewer_X7he · 2026-03-06

**Soundness:** 3
**Presentation:** 2
**Significance:** 3
**Originality:** 3
**Overall Recommendation:** 4
**Confidence:** 3

**Summary:**

The paper proposes a framework for AI alignment, pivoting away from the traditional goal of aligning a single isolated model. Instead, the authors model alignment as an emergent property of a competitive market of differently misaligned AI agents. Formulating the interaction as a multi-leader Stackelberg game, the authors introduce the "approximate market alignment" condition: the assumption that a human user's utility function lies approximately within the convex hull of the competing AIs' utility functions.

The authors provide theoretical guarantees demonstrating that if this condition holds, a user can achieve an outcome comparable to interacting with a single perfectly aligned model. They prove this across three interaction protocols: a fully strategic game, a bounded-rationality model using quantal response, and a simplified "Best-AI Selection" game. Finally, they offer empirical validation using prompt-perturbed LLMs on the ETHICS and MovieLens datasets, as well as real-world polling data from OpinionQA, to show that the convex hull of diverse AI models consistently captures human preferences better than any individual model.

**Compliance With Llm Reviewing Policy:**

Affirmed.

**Final Justification:**

The rebuttal has addressed my main concerns about the limitations of the proposed methodology, therefore, I updated my score. Once the proposed clarifications in the main text are done, it will significantly improve the readability and clarity of their contributions.

**Key Questions For Authors:**

1. How would the market alignment condition fare if the competing models suffer from correlated failures or biases (e.g., standard RLHF mode collapse)? Have you run any tests combining entirely different foundation models (e.g., Claude, Gemini, Llama) rather than relying on prompt perturbations of a single model?
2. In the Best-AI Selection Game, Alice must compute the expected utility over the prior distribution for each deployed model. What are the practical implications if Alice's evaluation set is small or heavily biased?
3. Could you elaborate briefly in the main text on how the "Information Substitutes" condition might break down in real-world scenarios where AI and human knowledge are strictly complementary?

**Limitations:**

yes

**Strengths And Weaknesses:**

*Strengths*
1. Novel Conceptual Framing: Approaching alignment via market competition and convex hulls rather than isolated optimization is a new perspective. It directly addresses the real-world scenario where multiple corporate LLM providers offer competing, uniquely biased models.
2. Strong Theoretical Foundation: The extension of multi-sender Bayesian persuasion to multi-round conversational protocols between differently informed parties provide a well-formulated framework. The progressive relaxation of assumptions from a fully strategic user to a boundedly rational one, and finally to a simplified selection game makes the theoretical claims highly robust.
3. Restrictive assumption relaxation: The authors propose relaxations to assumptions, e.g., bounded rationality instead of perfect rationality and commitment to a single provider after evaluating multiple option.s

*Weaknesses*
1. Ecological Validity of the AI Generation: In the ETHICS and MovieLens experiments, the authors simulate "competing AIs" by prompting a single underlying model to adopt slightly different personas or rephrased instructions. This may artificially inflate the diversity of the utility space. In reality, commercial models from different providers (OpenAI, Google, Anthropic) might share similar underlying alignment biases (e.g., universal safety filters), meaning their convex hull might not smoothly cover the true human utility space. The claim that any utility function lies in the convex hull of common would require a more realistic evaluation with querying common APIs.
2. Complexity of the Human Strategy: The foundational theorem (Theorem 3.4) requires the "Identical Induced Distribution Condition", meaning the human user must be sophisticated enough to extract and act upon their Bayes-optimal action regardless of which AI provides the information. While the authors elegantly address this with the quantal response model, the cognitive burden placed on the human to correctly aggregate these signals remains high. This counteracts the motivation of the work that states that the more AI providers are on the market the more likely a user's utility function is within their convex hull as this cognitive (and monetary) burden grows with the number of providers.
3. Writing: There are several concerns about the writing that makes understanding the paper challenging. For example, the paper extensively discusses results in the main text's Introduction that are later deferred to the Appendix (See bullet points in Section 1.1). A detailed example would further help understanding of conversation protocols, world states, features, and Alice's action.

---

> ### Author Rebuttal · Authors · 2026-03-30
>
> We thank the reviewer for the detailed feedback. We address each concern below.
>
> **W1: Ecological validity of the AI generation**. Our experiments are broader than they may appear from the main text. Three points:
> 1. The OpinionQA experiments (Section 4; full details in Appendix J.3) use responses from 9 different commercial LLMs, not prompt variations of a single model. The convex hull of these real model opinions aligns substantially better with human opinions than any individual model (Figure 5). This directly tests whether the assumption holds for a pool of distinct commercial models.
> 2. Beyond prompt perturbations, Section J.2 (Figure 4) tests alignment using fundamentally different evaluation perspectives e.g., "utilitarian philosopher," "legal analyst," "social justice advocate" for ETHICS, and genre-specific fans for MovieLens. Results are consistent with the perturbation experiments.
> 3. The perturbation setup tests the story from our introduction and Appendix D: if each AI developer tries to build an aligned model but produces a noisy approximation, the resulting utility functions are random perturbations of the target. Proposition D.1 shows that with enough such noisy attempts, market alignment holds with high probability.
>
>
>
> **W2: Complexity of the human strategy.** We agree this is an important point. The cognitive burden on Alice actually decreases across our three results:
> - Theorem 3.4 (Section 3): Alice must reason strategically about how to interact with the AIs. This is the most demanding setting.
> - Section G (Quantal Response): Alice follows a simple fixed rule, she shares her current beliefs honestly and picks actions roughly in proportion to how good they look. No strategic optimization required.
> - Theorem H.3 (Best-AI Selection): Alice simply tries out each model, picks the one that works best for her, and sticks with it. She only interacts with one model. This is what users already do when choosing between ChatGPT, Claude, Gemini, Grok, etc.
> Note that our results can be modified so that Alice is not required to optimize over a large strategy space. If she is limited to a small set of simple interaction strategies, the guarantees still hold, as long as she picks the best strategy available to her within that set.
>
> **W3: Writing and organization.** In the revision, we will: (1) shorten the discussion of results 2 and 3 in Section 1.1 to just the key takeaway for each, (2) add a concrete running example that walks through the conversation protocol, world states, features, and Alice's action, and (3) add high-level proof sketches for the main theorems. We will also better signpost how the three results progressively relax assumptions.
>
> **Q1. Correlated failures.** We agree that this can be a concern for the validity of market alignment condition. If models share biases (e.g., from similar RLHF training), the convex hull they span will be smaller, making $\varepsilon$ larger and weakening our guarantees. Our bounds ($OPT − 2\varepsilon$) make this degradation explicit, they get worse in direct proportion to how similar the models are. The OpinionQA experiments use 9 distinct commercial LLMs and still show meaningful convex-hull improvements, suggesting current model diversity already provides useful coverage.
>
> **Q2: Small or biased evaluation sets in Best-AI Selection.** If Alice evaluates models on a limited number of examples, she might pick the wrong one. The guarantee degrades gracefully: if her utility estimates are accurate to within $\eta$, the model she picks is at most $2\eta$ worse than the truly best one, so the Section H bound weakens by an additional $2\eta$. If the evaluation set is biased (not representative of the true distribution), the guarantee holds with respect to that evaluation distribution instead. We will state this in the revision.
>
> **Q3. Information substitutes.** This assumption is only needed for our second, quantal-response result in Appendix G. Our first result (Section 3) and the Best-AI Selection result (Section H) do not require it. The condition says, roughly, that Alice's information and the AI's information overlap in what they predict, knowing one makes the other less important. It fails when the two sources are strongly complementary: for example, if Alice holds one half of an encrypted message and the AI holds the other half, neither can learn anything alone, but together they can read the message. Many human-AI scenarios plausibly satisfy this condition. For instance, if both a doctor and a medical AI observe noisy signals about the same patient, their observations overlap and are naturally substitutable. We will clarify the scope of this assumption in the main text.

---

> > ### Author Rebuttal · Reviewer_X7he · 2026-04-02
> >
> > The authors answered my questions which improved my understanding of the paper and the limitations of the proposed method. I reassessed my evaluation accordingly and there are no further question.

---

### Official Review · Reviewer_tjWd · 2026-03-12

**Soundness:** 2
**Presentation:** 3
**Significance:** 3
**Originality:** 3
**Overall Recommendation:** 4
**Confidence:** 4

**Summary:**

The paper studies the question of aligning AI systems with human values by framing the problem as a multi-leader Stackelberg game, extending the Bayesian persuasion information design model. A key assumption of the work is the market alignment assumption, which means that the user's utility can be approximated as a weighted combination of the AIs' utilities.

Theoretically, the authors show that under a specific ‘no-gap’ assumption, the user can achieve their first-best utility in equilibrium. They also extend this result to the quantal response case. Furthermore, they introduce a modified setting that drops the 'no-gap' requirement, showing that the user can achieve an equilibrium utility approximately equal to interacting with a single perfectly aligned model.

Empirically, the authors provide validation for both the market alignment assumption and their theoretical utility guarantees using simulations with synthetic AI utility functions

**Compliance With Llm Reviewing Policy:**

Affirmed.

**Final Justification:**

My questions have been resolved, and I have no further concerns. I will maintain my positive score.

**Key Questions For Authors:**

- My understanding is that the user does not need to know the weights in the market alignment assumption in order to best respond or choose the best AI. Is this correct?

**Limitations:**

yes

**Strengths And Weaknesses:**

Strengths
1. It is an interesting way to frame the AI system alignment problem as a multi-leader Stackelberg game. The presentation is clear and easy to read.
2. Intuitively, the market alignment assumption is reasonable and becomes easier to satisfy as the size of the AI system grows. The authors also provide empirical justification for this.
3. The theoretical results are well organized: they cover idealized scenarios under the 'no-gap' assumption, bounded-rationality settings under a weaker 'no-gap' assumption, and best-AI selection cases without the 'no-gap' assumption.


Weaknesses
1. The inclusion of features is not well motivated. The theoretical results appear to hold even without them, and the empirical results do not clearly demonstrate how the features are designed or used.
2. Figure 1 should be explained more clearly. For example, I assume that NNLS refers to non-negative least squares, but this is not defined in the text. Similarly, the other weight-combination methods shown in Figures 1(a) and 1(b) should also be clearly explained.
3. The construction of the AI utility functions and the choice of dataset are not well motivated, which makes the empirical results somewhat narrow and limits the robustness of the results.

---

> ### Author Rebuttal · Authors · 2026-03-30
>
> We thank the reviewer for the careful read and constructive suggestions.
>
> **W1: Features are not well motivated.** Features are not an optional addition—they are what make conversation useful. $x_A$ and $x_B$ represent private information held by the user and AI models. In our medical diagnosis scenario: $x_A$ might be symptoms the doctor observes, $x_B$ might be patterns the AI has extracted from medical literature, and $y$ is the true diagnosis. The whole point of conversation is for Alice and the AIs to exchange this private information so Alice can make a better decision. Without private information on the AI side, there would be nothing for Alice to learn from conversation. While the convex-hull inequalities themselves do not explicitly depend on features, the benchmark they guarantee, Alice's utility when interacting with a perfectly aligned model, depends directly on what information is available $(x_A, x_B, y)$. Features are also needed for the results in Section G, where the Information Substitutes condition governs how well Alice can learn from conversation. We will add this motivation to the revision.
>
> **W2: Figure 1 should be explained more clearly.** Agreed. NNLS stands for Non-Negative Least Squares: it finds the best non-negative weighted combination of the Bobs' utilities to approximate the Alice’s utility. "Simplex" adds the constraint that the weights sum to 1 (a convex combination). "Best Individual" is the single Bob closest to Alice. "Simple Average" uses equal weights. In the revision, we will define these explicitly in the main text and expand the Figure 1 captions.
>
> **W3: Empirical construction and dataset choice**. Our goal is to test whether the convex hull of misaligned models can approximate a user's preferences better than any single model. We chose three domains that test this in different ways: moral judgments (ETHICS), recommendations with real human ratings as ground truth (MovieLens), and real human/LLM opinion data from 9 distinct commercial models (OpinionQA). Two details we will make more prominent: (1) Appendix J.2 tests with semantically different personas (e.g., "utilitarian philosopher," "legal analyst," "social justice advocate" and not just prompt rephrasings), and (2) Appendix J.3 uses actual opinion distributions from 9 different commercial LLMs (not variations of a single model) on real-world polling data.
>
> **Q1. Weights in the market-alignment assumption.** Yes, your understanding is correct. The weights are only used in the analysis. Alice’s best response depends only on her own posterior expected utilities, and in the Best-AI Selection game she only evaluates which deployed model gives her highest expected utility. Neither the user nor the providers need to know the convex-hull coefficients.

---

> > ### Author Rebuttal · Reviewer_tjWd · 2026-04-01
> >
> > Thank you to the authors for their responses. My questions have been resolved, and I have no further concerns. I will maintain my positive score.

---

### Official Review · Reviewer_6oHe · 2026-03-13

**Soundness:** 4
**Presentation:** 4
**Significance:** 3
**Originality:** 4
**Overall Recommendation:** 5
**Confidence:** 4

**Summary:**

This paper tackles the problem where a user might interact with misaligned AI systems. Through the framework of Bayesian persuasion in a multi-leader Stackelberg game, it shows one overarching finding in three different variants and settings of the game: A user can determine an (approximately) optimal output (with regards to its own preferences) from the outputs of multiple misaligned AIs as long as the users utility function lies in the convex hull of the AIs utility functions. With more AI systems, they argue, this condition will be more likely to be satisfied, and they show some experiments along those lines in a stylized setups with three datasets.

**Compliance With Llm Reviewing Policy:**

Affirmed.

**Final Justification:**

I maintain my positive assessment of the paper

**Key Questions For Authors:**

None at this point.
This should not be confused with a disinterest in the paper, I liked it.

**Limitations:**

Yes

**Strengths And Weaknesses:**

I advocate for its acceptance.

Strengths:
- It studies a significant problem in a fitting formal framework, and is able to derive results with a clear interpretation
- It is well-written with a great overview of results in Section 1

Neutral:
While the experiments are rather simple in design, they cover a range of variation one might encounter.

---

> ### Author Rebuttal · Authors · 2026-03-30
>
> We thank the reviewer for the very positive assessment and support. We are glad that the core framing, the interpretation of the results, and the overview in Section 1 came across clearly. We appreciate the encouraging feedback.

---

> > ### Author Rebuttal · Reviewer_6oHe · 2026-04-01
> >
> > I appreciate the contribution of this paper

---

### Official Review · Reviewer_1L4g · 2026-03-13

**Soundness:** 3
**Presentation:** 4
**Significance:** 3
**Originality:** 4
**Overall Recommendation:** 5
**Confidence:** 3

**Summary:**

This paper studies a setting in which a single user interacts with multiple AI agents with misaligned utilities, in the presence of an underlying state that is unknown to both the user and the agents. After engaging in conversations with the agents, the user makes a decision to maximize her own utility. By modeling this setting as a multi-leader Stackelberg game with multiple rounds of Bayesian persuasion, the authors derive three main results under the assumption that the user’s utility lies in the convex hull of the agents’ utilities. They also provide empirical results that support their theoretical arguments.

**Compliance With Llm Reviewing Policy:**

Affirmed.

**Final Justification:**

Thank the authors for their effort. I will maintain my positive score.

**Key Questions For Authors:**

I am interested in the following questions and would appreciate it if the authors could provide some intuition:
1. In practice, there are likely to be many users rather than a single representative user, and these users may have different or even conflicting utilities. If the game were extended to include multiple users, what would be the main barriers to deriving an equilibrium? Do your current results offer any insights into this more general setting?
2. Is there any impossibility result once the convex-hull assumption is removed?
3. What happens if a subset of the AI agents can collude or share information? In such a setting, could an equilibrium violate the guarantee even when convex-hull coverage still holds?

**Limitations:**

Yes.

**Strengths And Weaknesses:**

### Strengths
1. Extending Bayesian persuasion to a multi-leader, multi-round conversational setting with differently informed leaders and a follower offers a novel perspective of modelling the alignment problem.
2. The paper presents both strong theoretical insights and compelling empirical results in the problem of AI alignment with competition.

### Weaknesses
1. The main concern is whether the convex-hull assumption is truly mild in this context. If the agents share common misaligned biases or can collude, the assumption and the results may become fragile.
2. I also feel that the paper is somewhat too long for an 8-page format, which makes it difficult to verify all of the claims from the main text alone, especially the second and third key results. Would be better if some high-level ideas can be provided in the main text.

---

> ### Author Rebuttal · Authors · 2026-03-30
>
> We thank the reviewer for the thoughtful and encouraging review.
>
> **W1: Is the convex-hull assumption truly mild?** We agree that the convex-hull condition is not free. If models share similar biases (e.g., all trained by the same company, or all fine-tuned with similar RLHF objectives), $\varepsilon$ can remain large. Our point is that this condition on the market as a whole is much weaker than requiring any single model to be well-aligned. The condition also gets easier to satisfy as more independently developed models enter the market, which we show both theoretically (Proposition D.1) and empirically (Figures 1a, 1b, 2, 5).
>
>
> **W2: Paper length / main-text intuition.** We agree that the second and third results (Sections G and H) are hard to verify from the main text alone. In the revision, we will add a high-level proof sketch for each result in the main text, focusing on the key ideas rather than full formalism.
>
> **Q1: Multiple users.** We agree this is an important next step. The main technical difficulty is not equilibrium existence per se, but that with heterogeneous users there is generally no single conversation rule $C_B^*$ that is simultaneously best for all users. Different users may require different convex-hull weights and different best-response conversation rules, while providers commit to one deployed rule. If users come in identifiable types and providers can condition on user identity, our proof appears amenable to extension user-by-user (or ex ante over a distribution of user types): one would require market alignment to hold for each type, potentially with type-specific weights. The harder case is one fixed rule serving many conflicting utilities without knowing which user is which; this connects to the pluralistic alignment literature and is briefly discussed in Appendix B. An encouraging observation from that discussion: because different users can respond differently to the same set of deployed models, a single set of AI conversation rules could benefit many users at once: each user extracts the information most useful to them.
>
> **Q2: Impossibility without the convex-hull assumption.** We do not prove a necessary-and-sufficient characterization, and we agree weaker sufficient conditions are possible (indeed Remark H.4 already weakens the assumption to only the outcome sets relevant for the proof). But some assumption of this flavor is necessary: if all providers share the same misaligned utility, the multi-provider game reduces to the single-provider misaligned case and competition cannot help. Our simulations (Figure 1c, Appendix K) further confirm this: when misalignment ε is large, Alice's equilibrium utility can be substantially below OPT. So our contribution is a clean sufficient condition that is substantially weaker than individual alignment, not a complete characterization.
>
> **Q3: Collusion / information sharing.** Information sharing by itself does not break the proofs: the arguments depend only on the final distributions over Alice's actions and world states, so richer shared information can be absorbed into the agents' conversation rules. For collusion, it depends on the framework: (i) Our guarantees hold for every Nash equilibrium, so they automatically hold for any refinement of Nash equilibrium, including coalition-proof ones. (ii) If several providers are effectively controlled by one actor with a common objective, then the relevant market-alignment condition should be stated over the independent strategic units (coalitions). This  could shrink the effective convex hull and increase $\varepsilon$. We will add a brief discussion of collusion robustness in the revision.

---

> > ### Author Rebuttal · Reviewer_1L4g · 2026-04-01
> >
> > Thank the authors for their reponses. My questions are resolved and I have no further questions. I will maintain my positive score.

---

### Decision · Program_Chairs · 2026-04-30

**Decision:**

Accept (regular)

**Comment:**

All reviewers agree that this paper offers a meaningful contribution to the field. It extends Bayesian persuasion between differently informed parties, and then offers strong theoretical insights and compelling empirical evidence of the methods success.  I recommend the paper be accepted to ICML